




# A Mathematical Framework for the Description of
# Convection in Meso-scale Synoptic System


Nan Zhao[1]

*State Key Laboratory of Severe Weather,*
*Chinese Academy of Meteorological Sciences, Beijing, 100081, China*


Masaaki Takahashi

*Atmosphere and Ocean Research Institute (AORI),*

*University of Tokyo, 5-1-5 Kashiwanoha, Kashiwa, 277-8568, Japan*


[1]*Corresponding author*: Dr. Nan Zhao, State Key Laboratory of Severe Weather, Chinese Academy of
Meteorological Sciences, Beijing, 100081, China. E-mail address : zhaon@cams.cma.gov.cn

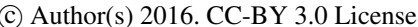

## Abstract
By introducing an appropriately-defined imbalanced vortical flow as the basic state, our previous
study has extended conventional instability theories of balanced flows for meso-scale convection. It
considered not only the apparent instability of the imbalanced basic state but also the two-way
interaction between convection/IGWs and this imbalance. This paper reports our *new progresses* of
such framework. A regular perturbation method on the nonlinear case is performed to have an insight
into the triggering mechanism of convection. It seems convection can be triggered in resonance either
with imbalance forcing or with nonlinear interaction among different modes. Even if all these cannot
happen, an imbalance forcing with strong enough magnitude may eventually trigger convection. These
are essentially different from the concept of Liyapunov instability, in which an initial disturbance is
necessary. In some simplified but relatively general setting, all modes that may contribute to the
structures of meso-scale convection are investigated, including free modes of convection and forced
modes of convection/IGWs by imbalance. Particularly, the influences of arbitrary distribution of
stratification on qualitative properties of free and forced convection/IGW modes are discussed. Also,
approximate forms of forced convection/IGW modes suitable for application are given for horizontally
uniform stratification. Finally, to demonstrate the potential application of our theory, the concept of
imbalance forcing and balanced flow adjustment is shown to be useful in the understanding of key
issues in typhoon study, such as its possible role in typhoon's self-organization, Fujiwhara effect and the
relationship between typhoon's asymmetric structure and its track recurvature.

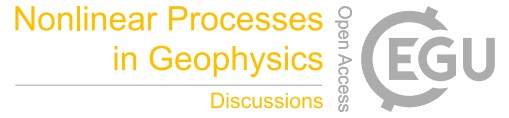

## 1 Introduction

Earlier theories attributed the arising of convection to the instabilities of balanced flows such as static instability, symmetric instability and etc. (Hoskins, 1974; Holton, 1992; Xu and Clark, 1985). However, many issues of convection still cannot be understood within the perspective of instability because of the following reasons. First, since classical theory of instability demands basic state must be a strict solution or exactly balanced flow, previous studies deal with too simple cases of balanced flows (mainly static state or parallel geostrophic flows with vertical/horizontal shears, see Pedlosky, 1979; Drazin, 1981 Holton, 1992) to have further applications. In real atmosphere, the instability theory for a general basic state which needs not to be a strict solution or balanced flow is necessary, not only because of the difficulties in ensuring the existence and finding out the exact solution for a general basic state, but also because of the highly imbalanced natures in synoptic systems of meso-scales. Second, meso-scale convection should be considered in the context of its two-way interaction with its basic state of larger scales (Emanuel et al., 1994; Roode et al., 2004). So, instability with the prescribed basic state as a strict solution of the nonlinear equations of motion cannot be a suitable description for such interaction as disturbance can never react on the basic state under this circumstance. Last, in the sense of classical Liyapunov stability, an extra initial disturbance is necessary for the triggering the instability, while the existence of such disturbance is hard to be identified in real atmosphere which is an ultimate state of long time evolution.

Our previous work which was motivated by the expectation to tidy these interconnected issues up, has incorporated meso-scale convective activities in the framework of instability problems of imbalanced basic state defined appropriately as an imbalanced vortical flow (Zhao, et al, 2011). Both loss of balance and loss of stability and their influences on the onset, development of convective activities had been investigated as a preliminary work. However, investigations on many key issues are still far from sufficient. Some important concepts need to be extended and clarified further and some key specific issues need to be discussed more sufficiently to enhance the framework. Emphases of this



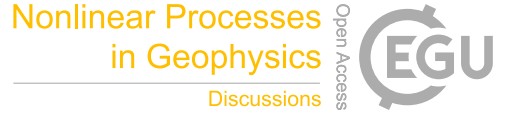

paper are placed on our new progresses in following three issues of convection of imbalanced basic
flows: 1) The triggering mechanism of convection, an key issue of convection that still remains unclear
partly because the mathematical nature of such issue is not well understood in our previous work. By
utilizing a perturbation analysis, an further insight into this issue is conducted and gives interesting new
results; 2) We know that free and forced modes of convection and inertia-gravity waves (hereafter,
referred as IGWs) are essential to understand the structure of convective activities, whereas the
distribution of stratification (stable or unstable) may affect greatly the behaviors of these modes. The
situation of horizontally uniform stratification had been considered in our previous work. However, no
mathematical way is found to cope with the situation of an arbitrarily distributed stratification so as to
draw some general conclusions on these modes. By proposing an linear eigenvalue problem in some
simplified but relatively general setting, we give qualitatively resolution to this issue as well as
interesting conclusions. We believe these linear modes may serve as the basis to understand the
structure of convective activities which is intrinsically nonlinear phenomenon that comprise interaction
among a multitude of such linear modes at various scales and is difficult to be dealt with. In addition,
we also need to modify the results of horizontally uniform stratification that we obtained previously and
transform them into some applicable forms for our subsequent study of application; 3) So far, we don't
have an successful example of application to demonstrate the usefulness of our previous theory, for
which we choose the study of typhoon properties such as its formation and the relationship between
typhoon recurvature and its asymmetric structure as the example.
The paper is arranged as follows. In section 2, we give a brief summary of the basic theory and
related concepts of our previous study, and provide the basic equations necessary for our further study
in this paper. In section 3, as an important aspect of two-way interaction of convection with its
environment, triggering mechanism of convection is investigated via a perturbation analysis. We try to
find qualitative properties for convection and IGW modes of some general cases in section 4, where a
arbitrarily distributed stratification with unstable zone included is assumed. In section 5, we consider the



special and solvable case of a horizontally uniform stratification with also unstable zone included is
assumed. Rather than only forced convection mode are considered as in Zhao, et al. (2011), all vertical
motions that may contribute to the structure of convection in meso-scale system are given, including
those caused by free modes of instabilities, forced convection and IGWs. The approximate forms that
are applicable for the straightforward estimation of convection/IGW structures inside and outside a
meso-scale system are also derived. In section 6, physical explanations of above results are given so as
to tie them with convective activities of various meso-scale synoptic systems in real atmosphere, which
may service as a brief direction for the potential application of our theory. Particularly, applications in
the explanation of typhoon's structure and motion are provided as an example of detailed study. Section
7 is devoted to a summary and conclusions of the whole paper.
**2 Basic concepts and related equations of the theory**
The basic equations applicable to meso-scale convection system in our previous study (Zhao, et al, 2011)
are vorticity equation, divergence equation and thermodynamic equation in *p*-coordinates and *f*-plane. For
the simplicity, the forms of these equations and their properties related to balanced flows are given in
Appendix A. Although hydrostatic assumption cannot give the most exact description of each single
small-scale cell of convection, it has the advantages to deal with the structure of meso-scale convective
activity and its interaction with the background. In this sense, this paper actually treat with meso-scale
convective activity. However, above form of basic equations is unable to provide us an intuitive basis for
both the mathematical and physical aspects of the relation between imbalanced basic flow and meso-scale
organized convective activity characterized by convection/IGWs modes. So, by reducing these basic
equations to just one equation, we introduce a new form of equation, in which imbalance, convection/IGWs
and their relation can be recognized more easily. This new equation leads

$$\frac{\partial^2 \delta_{pp}}{\partial t^2} + \sigma \nabla^2 \delta + f^2 \delta_{pp} + \Im(\delta) - \ell_{\varsigma,\varphi}\delta = \Re(\varsigma,\varphi) \qquad (1)$$

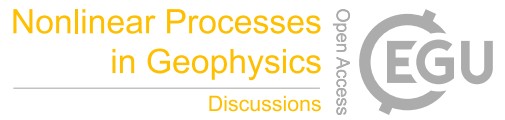

1    Where $\delta$ is the horizontal divergence, $\varsigma$ the vertical component of vorticity, $f$ the Coriolis-parameter

2    and $\sigma$ the static stability parameter. The linear and nonlinear operators acting on $\delta$, *i.e.* $\ell_{\varsigma,\varphi}\delta$ and

3    $\Im(\delta)$ are given as below

$$\ell_{\varsigma,\varphi}\delta = [-f\nabla\varsigma\cdot\mathbf{V}_\delta - f\frac{\partial\varsigma}{\partial p}\omega - f\varsigma\delta + f\mathbf{k}\cdot(\frac{\partial\mathbf{V}_\varsigma}{\partial p}\times\nabla\omega)]_{pp}$$

$$-[\mathbf{V}_\varsigma\cdot\nabla\delta + (a_\varsigma a_\delta + b_\varsigma b_\delta) + \frac{\partial\mathbf{V}_\varsigma}{\partial p}\cdot\nabla\omega]_{tpp} + \nabla^2[\nabla(\frac{\partial\varphi}{\partial p})\cdot\mathbf{V}_\delta]_p \qquad (2a)$$

$$\Im(\delta) = f[-\mathbf{k}\cdot(\frac{\partial\mathbf{V}_\delta}{\partial p}\times\nabla\omega)]_{pp} - (\mathbf{V}_\delta\cdot\nabla\delta + \omega\frac{\partial\delta}{\partial p} + \frac{\partial\mathbf{V}_\delta}{\partial p}\cdot\nabla\omega)_{ppt}$$

$$-\frac{1}{2}(\delta^2 + a_\delta^2 + b_\delta^2)_{ppt} \qquad (2b)$$

Here, *V* is the horizontal wind with zonal component $u$ and meridional component $v$, $\omega$ the vertical wind
and $\varphi$ the geo-potential height. $a = \partial u/\partial x - \partial v/\partial y$ and $b = \partial v/\partial x + \partial u/\partial y$ are deformations of the
horizontal wind. The horizontal wind speed can be decomposed into vortical and divergent parts, *i.e.*,
$\mathbf{V} = \mathbf{V}_\varsigma + \mathbf{V}_\delta$ , where the mean or constant part of the wind speed is absorb into $\mathbf{V}_\varsigma$. The subscript $t$ and $p$
denote the partial derivative with respect to them. On the right-hand side of (1),

$$\Re(\varsigma,\varphi) = -\frac{1}{2}[(a_\varsigma^2 + b_\varsigma^2 - \varsigma^2)_t]_{pp} - f(\mathbf{V}_\varsigma\cdot\nabla\varsigma)_{pp} + \nabla^2[\mathbf{V}_\varsigma\cdot\nabla(\frac{\partial\varphi}{\partial p})]_p \qquad (3)$$

is the inhomogeneous term which depends only on $(\varsigma,\varphi)$. Since vorticity equation, divergence equation
and thermodynamic equation are substituted into (1) in the derivation, constraints form basic dynamical
and thermo-dynamical laws that one might expect in convective systems in the real atmosphere still work
to a certain extent in (1). Nevertheless, equation (1) is more appropriately to be regarded as a diagnostic
equation for the qualitative relationship between imbalanced basic flow and convection/IGWs. Although it



alone is not closed and cannot serve as the governing equation to determine the motion, it does serve as one
of the constraints for the motion. In another words, it may unable to describes all aspects of the motion, but
it can describes qualitatively one aspect: the relationship between unbalanced basic flow and
convection/IGWs.
The balanced flow is a purely vortical flow with $\delta=0$ and is delineated just by $(\varsigma,\varphi)$. Obviously,
staticstate, parallel geostrophic flow and axisymmetric gradient flow are just particular cases of balanced
flows. One of the key points of our previous work is that we decompose the phase state $S$ of the dynamical
system as below

$$\mathbf{S} \equiv \begin{pmatrix} \delta \\ \varsigma \\ \varphi \end{pmatrix} = \begin{pmatrix} 0 \\ \varsigma \\ \varphi \end{pmatrix} + \begin{pmatrix} \delta \\ 0 \\ 0 \end{pmatrix} \equiv \mathbf{S}_0 + \mathbf{S}'$$

(4)

where $S_0$ is the basic state and $S'$ the disturbance. In other words, $(\varsigma,\varphi)$, the vortical component $\varsigma$ together
with $\varphi$ is viewed as a generalized basic state, regardless whether or not they are exactly balanced, while
the divergent component $\delta$ is the disturbances about it.
It is proven that, if the basic state $(\varsigma,\varphi)$ is an exactly balanced flow or an exact solution, we have
$\Re(\varsigma,\varphi) = 0$, then (1) returns to the original problem of instabilities including static instability and
symmetric instability *etc*. The corresponding types of linear instabilities with $\Im(\delta)$ omitted are given in
Table I. However, the exactly balanced flows are just particular cases. In a meso-scale system, the basic
state $(\varsigma,\varphi)$ may remains far apart from balance or even no such solution of balanced flow can exist.
Under such circumstances the inhomogeneous term remains $\Re(\varsigma,\varphi) \neq 0$ in (1), which appears as some
external forcing of the basic state on convection. As a result, in addition to producing instabilities, the role
of imbalanced basic flow on convection seems also to be a forcing by its imbalance. These are even more
clearly seen from the quasi-linear version of (1) without $\Im(\delta)$, namely

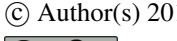

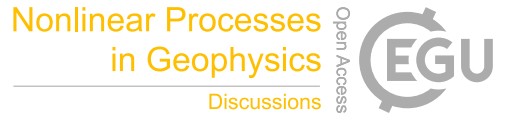

$$\frac{\partial^2 \delta_{pp}}{\partial t^2} + \sigma \nabla^2 \delta + f^2 \delta_{pp} - \ell_{\varsigma,\varphi} \delta = \Re(\varsigma, \varphi) \qquad (5)$$

Its solution is the superposition of homogeneous solution $\delta_h$ and inhomogeneous solution $\delta_i$, *i.e.*

$$\delta = \delta_h + \delta_i \qquad (6)$$

The homogeneous part of (5) and its solution $\delta_h$ behave like a problem of linear instability of the basic
state $(\varsigma, \varphi)$, even if $(\varsigma, \varphi)$ is not an exactly balanced flow having $\Re(\varsigma, \varphi) \neq 0$, which is called by us an
*apparent* instability because the original concept of stability/instability is for balanced flow or exact
solution. Meso-scale convection is related with this kind of apparent instability having nearly the form of
static instability, Kelvin-Helmholtz instability, inertia instability or symmetric instability *etc*. The
inhomogeneous solution $\delta_i$ of (5) for an unstable homogeneous operator is different from that of a stable
one of forced IGWs or spontaneous emission (Lighthill 1952; Ford, 2002). So both the homogeneous and
the inhomogeneous solution contribute to the meso-scale convection. Consequently, rather than the
traditional way which defines convection as the vertical motion arising from the instabilities of exactly
balanced flows, the generalized definition regards convection as the results of both apparent instability and
the response to the forcing of an imbalanced basic flows with such apparent instability. On the whole, by
using a quasi-linear version of the basic equation, we can decouple instability and response to imbalance.
As a result, smaller scale of instability and larger scale imbalance of $(\zeta, \varphi)$ are separated as well. As $(\zeta, \varphi)$
and its imbalanced is mainly at meso-scale, it can serve as a basic flow of the convective activities.
**3 Triggering of convection and its interaction with basic flow**

24        The characters concerning interaction between convection and its basic imbalanced vortical flow depend

highly on development stages of convection (Tao *et al*, 1979). Characters of such interaction for *developed*


stage of convection are not the whole story. The *triggering* mechanism of the convection is another aspect
that constitutes the issue of two-way interaction. Both of these two aspects will be discussed in this section.
*3.1 Two-way interaction*: *developed convection*

4       Equation (1) actually describes implicitly a two-way interaction between convective activities and the

basic flow as well. Generally speaking, the reaction or feedback of the convection on the basic flow seems
far more complex to be described in the full nonlinear form of $\delta$. For the simplicity, we just consider the
quasi-linear case of this issue given by (5). But this does not need to mean the discussion is in linear sense,
because $\ell_{\varsigma,\varphi}\delta$ and $\Re(\varsigma,\varphi)$ are nonlinear terms. Under this circumstance, it is clear that 1) the free
convection given by homogeneous solution cannot react on the basic flow, and 2) convective activities do
act on the basic flow via forced convection and contribute to its adjustment, which is described by the
reconciliation between the left and right hand sides of (5), although such two-ways interaction can never be
dealt with in the framework of dynamical instabilities. In the present study we would like to address this
issue mathematically as below.
According to the Fredholm alternative (see any text book on partial differential equation, e.g. Haberman,
2003), the solvability of (5) requires its inhomogeneous term $\Re(\varsigma,\varphi)$ to be orthogonal to the homogeneous
solution $\delta_h$, namely
$$<\delta_h, \Re(\varsigma,\varphi)> = \int_\Omega \delta_h^* \Re(\varsigma,\varphi) d\Omega = 0 \qquad (7)$$
where $<,>$ is an inner product properly defined over some spatiotemporal domain $\Omega$, This constraint on
$\Re(\varsigma,\varphi)$ means that the imbalance is limited to some certain ways. We suppose the physical meaning of this
reaction of forced $\delta$ on $(\zeta,\varphi)$ is related to balanced flow adjustment, which tends to remove the imbalance
of $(\zeta,\varphi)$ and is the foundation of our application study in this paper.
The (approximate) satisfaction of solvability condition (7) also implies equation (1) can be approximated
by its quasi-linear form (5). So when (7) is even not satisfied approximately, it means that: 1) a resonance





between convection and imbalance basic flow happens and can be explained as another kind of manner of
this interaction; 2) equation (1) can no longer be approximated by (5) and a fully nonlinear form of $\delta$ is
needed, which may corresponding to strong convection.
*3.2 Triggering mechanism of meso-scale convection*

6        The triggering mechanisms of convection for balanced and imbalanced basic state are different. In the

balanced case, convection or instability grows from some initial disturbance about the state and is called as
a disturbance-triggered convection. In the imbalanced case, convection is a result of both apparent
instability and response to imbalanced forcing. In the linear regime of the development of convection, this
imbalance-forced part of convection ($\delta_i$) cannot interact with free unstable modes of apparent instability
($\delta_h$). If there is not an initial disturbance for $\delta_h$, $\delta_h = 0$ is always a solution of the homogeneous equation
and instability can never arise. So an initial disturbance for $\delta_h$ is also necessary for the development of
apparent instability, which is similar to the case of balanced vortical flows and can also be called a
disturbance-triggered convection. In addition to the old question of where this initial disturbance comes
from, another question leads us to doubt the disturbance-triggered mechanism of convection in real
atmosphere is that: instability should develops immediately once the atmosphere become unstable, while it
seems instability does not raise until some forcing of uplift strong enough happens in observation.
Therefore, theory of disturbance-triggered instability for balanced flow or linear regime of imbalanced flow
need to be improved to explain the triggering of convection. For this purpose, a nonlinear theory for
imbalanced flow is necessary as below.

21        Suppose imbalance around a balanced flow ($\zeta_0, \varphi_0$) is weak in equation (1), we then have

23                                        $$\Re(\varsigma, \varphi) \propto \varepsilon \qquad (8)$$



1   where $\varepsilon \ll 1$ is a small dimensionless number. Notice also that

$$\ell_{\varsigma,\varphi} = \ell_{\varsigma_0,\varphi_0} + \ell^1_{\varsigma,\varphi}$$
(9)

3   *i.e.* operator $\ell_{\varsigma,\varphi}$ can be divided into a balanced and an imbalanced parts. It can also be shown

$$\ell^1_{\varsigma,\varphi} \propto \varepsilon$$
(10)

5   The perturbation solution of equation (1) is written as

$$\delta = \delta_0 + \delta_1\varepsilon + \delta_2\varepsilon^2 + \cdots$$
(11)

Substituting it into (1), we have

9       $\varepsilon^0$:

$$L(\delta_0) - \Im(\delta_0) = 0$$
(12)

11      $\varepsilon^1$:

$$L(\delta_1) = \Re(\varsigma, \varphi) + \ell^1_{\varsigma,\varphi}\delta_0 + \Im_1(\delta_0)$$
(13)

13      $\varepsilon^2$:

$$L(\delta_2) = \ell^1_{\varsigma,\varphi}\delta_1 + \Im_2(\delta_0, \delta_1)$$
(14)

$$\cdots\cdots$$

16      $\varepsilon^n$:

$$L(\delta_n) = \ell^1_{\varsigma,\varphi}\delta_{n-1} + \Im_i(\delta_0, \delta_1, \cdots, \delta_{n-1})$$
(15)

where we denote





$$L(\cdot) = \frac{\partial^2 (\cdot)_{pp}}{\partial t^2} + \sigma \nabla^2 (\cdot) + f^2 (\cdot)_{pp} - \ell_{\varsigma_0, \varphi_0} (\cdot) \qquad (16)$$

Solution of equation at order $\varepsilon^n$ is a superposition of homogeneous and inhomogeneous solutions, *i.e.*

$$\delta_n = \delta_{nh} + \delta_{ni} \qquad (17)$$

And it is easy to show $\delta_0 = 0$ is a solution of the first equation. So, we have

$$\begin{aligned} \delta &= \delta_1 \varepsilon + \delta_2 \varepsilon^2 + \cdots \\ &= (\delta_{1h} + \delta_{1i})\varepsilon + (\delta_{2h} + \delta_{2i})\varepsilon^2 + \cdots \\ &= \delta_{ist} + \delta_{frc} \end{aligned} \qquad (18)$$

where

$$\delta_{ist} = \delta_{1h}\varepsilon + \delta_{2h}\varepsilon^2 + \cdots \qquad (19a)$$
$$\delta_{frc} = \delta_{1i}\varepsilon + \delta_{2i}\varepsilon^2 + \cdots \qquad (19b)$$

are defined according to their governing equations as the instability and response to imbalance,
respectively. This superposition relation is similar to that of linear case in (6). If at some initial time we
have $\delta_{ist}|_{t=0} \neq 0$, instability can be triggered, which is the situation of disturbance-triggered mechanism.
Otherwise, if at some initial time $t = 0$ we have

$$\delta_{ist}\big|_{t=0} = \delta_{1h}\big|_{t=0} = \delta_{2h}\big|_{t=0} = \cdots = 0 \qquad (20)$$

then $\delta_{ist} \equiv 0$. In the latter case, once forced solution

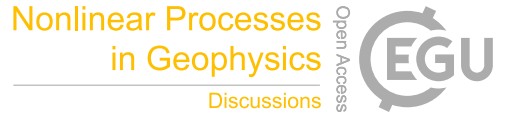

$$\delta_{frc} = \delta_{1i}\varepsilon + \delta_{2i}\varepsilon^2 + \cdots \qquad (21)$$

3 converges and exists, instability cannot be triggered. Conditions for solvability, according to the

4 Fredholm alternative, should be

$$\text{i)} \quad \int_{\Omega} \Delta^* \Re(\varsigma, \varphi)d\Omega = 0 \qquad (22a)$$

$$\text{ii)} \quad \int_{\Omega} \Delta^* [\ell^1_{\varsigma,\varphi}\delta_{n-1} + \Im_i(0, \delta_1, \cdots, \delta_{n-1})]d\Omega = 0 \qquad (22b)$$

$$\text{iii)} \quad \delta_{1i}\varepsilon + \delta_{2i}\varepsilon^2 + \cdots \quad \text{converges} \qquad (22c)$$

where $\Delta$ is the homogeneous solution of $L(\Delta) = 0$, while $\Delta^*$ is its adjoint solution. On the contrary,
when one of above conditions is not satisfied, instability arises. In fact, once solution $\delta_{frc}$ does not exist,
it implies solution $\delta_{ist} = 0$ can exist neither. So we begin to have $\delta_{ist} \neq 0$ and instability can thus arises
under unstable situation. Here, unsatisfication of i) and ii) may be explained as convection's resonance
for forcing from imbalance and nonlinear interaction among IGW modes, respectively. The phrase
"resonance" here seems too intangible to be related to the real problem. So we explain it in another ways
as below. Condition i) and ii) are explained as orthogonal relations between free mode of convection
and imbalance forcing or forcing by nonlinear interaction of IGW modes, they can further be explained
as no resemblance, or no correlation between the two space-time fields in the sense of statistics. As a
result, "resonance" or unsatisfication of i) and ii) can be explained as caused by some resemblance, or
statistical correlation between the space-time structures of free mode of convection and forcing either
from imbalance or from nonlinear interaction among IGW modes. Even if no such resonances can
happen, point iii) suggests the magnitude of imbalance (characterized by $\varepsilon$) be influential. As $\varepsilon$-order



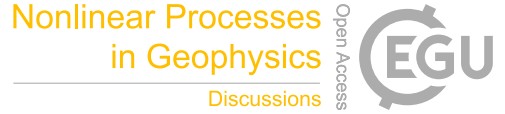

imbalance becomes strong enough, there may be a critical value $\varepsilon_c$. When $\varepsilon < \varepsilon_c$, solution $\delta_{frc}$ exists
together with $\delta_{ist} = 0$ and convection cannot arise. While once $\varepsilon > \varepsilon_c$, solution $\delta_{frc}$ can no longer exist
and also $\delta_{ist} \neq 0$, convection then arises. This explains well the time delay of the development of
convection after the atmosphere becomes unstable. The particular case of triggering mechanism of
convection for balanced flow without meso-scale synoptic systems accompanied, such as daytime
heating on a flat and homogeneous surface, can also be interpreted by these results. In this case we have
both $\Re(\varsigma, \varphi) = 0$ and $\ell_{\varsigma, \varphi} = 0$, so condition i) is satisfied automatically. If there are IGWs traveling
through the region considered from outside and condition ii) is not satisfied, the nonlinear interaction
among IGW modes may trigger unstable convection modes. A simple analysis as below is helpful for
the imaging of this situation. If $\sigma$ is $p$-dependent, suppose the linear IGW mode is $P_I(p)e^{-i\omega t}e^{ikx}$, it
produces a forcing with a portion of $P_I^2(p)e^{-i2\omega t}e^{i2kx}$ due to quadric nonlinearity. It is easy to show
this portion of forcing cannot be orthogonal to the unstable linear normal mode of convection
$P_C(p)e^{\lambda t}e^{i2kx}$, as long as vertical $P_C(p)$ is not orthogonal to $P_I^2(p)$, which is usually true because
vertical mode $P_C(p)$ is already orthogonal to $P_I(p)$ and can no longer be orthogonal to its
square. This case makes the physical meaning of point ii) even clear, regardless of IGWs is from outside
or generated by meso-scale imbalance itself. This is also consistent with our knowledge before. It has
been long for people to know IGWs is a triggering mechanism for convection in atmosphere with
conditional unstable stratification (see, e.g. Li, 1978). Although there are different explanations for this
fact, resonance between convection and nonlinear interaction among IGWs may be another possible. On
the other hand, unsatisfication of either i) or iii) indicates the role of imbalance forcing in triggering
convection via its structure or magnitude, respectively. This is also well-known fact by forecasters and
referred to imbalance forcing crudely as an uplifting effect by meso-scale circulation, without
distinguishing between the two cases.
For this triggering mechanism of the apparent instability, an external initial disturbance is not

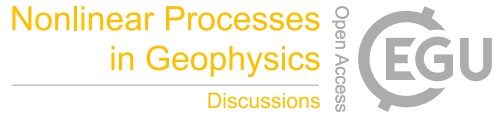
necessary, because we always have an initial disturbance $\delta_{ist}\mid_{t=0}\neq 0$ due to the nonexistence of forced
solution $\delta_{frc}$ for the imbalanced vortical flow. Then, imbalance can provide also an initial disturbance
from which the apparent instability or convection can develop. So, the triggering mechanism of
convection is attributed to the imbalance of the basic state rather than to some initial disturbance from
outside. We call this kind of convection without external initial disturbance a spontaneous convection.
A most important difference between the properties of disturbance-triggered and spontaneous
convection is that the former develops immediately no matter how small and what kind of structure the
initial disturbance may be, while the triggering and development of the latter depends highly on the
structure and strength of the imbalance and nonlinearity.
**4 Some general properties of simplified meso-scale convection system**
As we have known in above, the relationship between convection and its meso-scale basic flow is
characterized both by a response to imbalanced forcing $\Re(\varsigma,\varphi)$ and by the apparent instability of the
basic flow associated with $\ell_{\varsigma,\varphi}\delta$. When imbalance departure from a balanced flow $(\varsigma_0,\varphi_0)$ is weak,
the above discussion (equation (13) with $\delta_0=0$) suggests the first order linear approximation of (1) can
be written as

$$\frac{\partial^2\delta_{pp}}{\partial t^2}+\sigma\nabla^2\delta+f^2\delta_{pp}-\ell_{\varsigma_0,\varphi_0}\delta=\Re(\varsigma,\varphi) \qquad (23)$$

There are two typical cases of the balanced flow $(\zeta_0,\varphi_0)$ in meso-scale system. The first one is
symmetric balanced flow along a variable $\alpha$, i.e. $\partial\varsigma_0/\partial\alpha=\partial\varphi_0/\partial\alpha=0$. Here $\alpha=x\in(-\infty,\infty)$ or $\alpha=\theta\in$
[0, 2$\pi$) correspond to balanced parallel geostrophic wind or concentric gradient wind. Suppose
disturbance $\delta$ in above is arbitrary. An average with respect to $\alpha$ over its whole domain can change (23)
into a problem of forced symmetric instability, namely



$$\frac{\partial^2 \overline{\delta}_{pp}}{\partial t^2} + A\overline{\delta}_{\beta\beta} + B\overline{\delta}_{\beta p} + C\overline{\delta}_{pp} + E\overline{\delta}_{\beta} + F\overline{\delta}_{p} = \overline{\mathfrak{R}}(\varsigma, \varphi)$$

(24)

where $\beta = y$ or $\beta = r$ for balanced parallel geostrophic wind and concentric gradient wind, respectively. And

coefficient from $A$ through $F$ depend on $(\zeta_0, \varphi_0)$. Then we have symmetric instability as the homogeneous

solution and inhomogeneous forced convection. Conditions for the symmetric instability are

$$q = 4AC - B^2 < 0 \quad when \quad A > 0$$

(25)

Since the steady form of the governing equation (24) under such conditions becomes hyperbolic, the

forced convection can no longer be estimated from the structure of forcing $\overline{\mathfrak{R}}(\varsigma, \varphi)$, unlike the elliptic

equation when the equation is symmetrically stable.

The second case is in the highly imbalanced situation that one cannot find a symmetric balanced flow

$(\zeta_0, \varphi_0)$ close to $(\zeta, \varphi)$ like above. Under this circumstance, it is convenient to choose static basic state $(\zeta_0$

$=0$, $\varphi_0 = const.$ horizontally) as the nearest balanced flow and have $\ell_{\varsigma_0, \varphi_0} = 0$, which yields

$$\frac{\partial^2 \delta_{pp}}{\partial t^2} + \sigma \nabla^2 \delta + f^2 \delta_{pp} = \mathfrak{R}(\varsigma, \varphi)$$

(26)

Its homogeneous part is equal to an issue of static instability (IGWs) when $\sigma < 0$ ($\sigma > 0$) in some domain

of atmosphere. This is a very useful setting in the application study in this paper. So in such a setting,

properties of convection/IGWs depend highly on the distribution of $\sigma$. If $\sigma$ is $p$-dependent only, its

forced solution becomes solvable in spite of its steady part being a hyperbolic equation, which was



discussed previously in Zhao et al (2011) and will be improved in detail in the next section. But if there
are regionally confined unstable stratification in meso-scale system, it seems difficult to solve (26). To
address this issue, this section will be focused on the relatively general case of $\sigma$ being both $p$- and
horizontal-dependent arbitrarily and just give some qualitative properties of convection and IGW
modes.
*4.1 Linear modes of convection/IGW of arbitrarily distributed stratification*

8        We assume both $p$- and horizontal-dependent $\sigma(x, y, p)$ in equation (26) in our following

discussion. First of all, homogeneous part of equation is essential because 1) it decides the free modes
of convection/IGW and 2) such modes together with the structure of inhomogeneous forcing jointly
decide the existence of the forced convection via conditions for solvability (22a) and decide the actual
structure of the forced convection via inhomogeneous solution if it exists. So the homogeneous solution
of (26) is the foundation for the present issues and can be obtained by method of separation of variables
written as $\delta_n = T_n(t)A_n(x, y, p)$, so we have

$$\frac{T_{ntt}}{T_n} + f^2 = -\frac{\sigma\nabla^2 A_n}{A_{npp}} = -\lambda_n \qquad (27)$$

which yields an eigenvalue problem as below

$$\sigma\nabla^2 A_n - \lambda_n \frac{\partial^2 A_n}{\partial p^2} = 0 \qquad (28)$$

while the temporal part $T_n$ satisfies

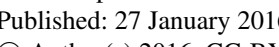



$$\frac{d^2 T_n}{dt^2} + (\lambda_n + f^2) T_n = 0$$

(29)

3 In the appendix B of this paper, it is proven that the real and imaginary parts of $\lambda_n$ are

$$\mathrm{Re}(\lambda_n) = \frac{\lambda_n + \lambda_n^*}{2} = \frac{\int_\Omega (\sigma \| \nabla A_n \|^2 - 1/2 \| A_n \|^2 \Delta \sigma) d\Omega}{\int_\Omega \| \partial A_n / \partial p \|^2 d\Omega}$$

(30a)

$$\mathrm{Im}(\lambda_n) = \frac{\lambda_n - \lambda_n^*}{2} = \frac{\int_\Omega \nabla \sigma \cdot (A_n^* \nabla A_n - A_n \nabla A_n^*) d\Omega}{2 \int_\Omega \| \partial A_n / \partial p \|^2 d\Omega}$$

(30b)

respectively. Here,

$$\int_\Omega (\cdot) d\Omega = \iint_S \int_0^{p_s} (\cdot) dp dS; \quad \Omega = S \times [0, \, p_s]$$

(31)

$S$ is the area of meso-scale system, while $p_s$ the surface pressure. By introducing an approximate
relation of $\sigma$

$$\Delta \sigma \approx -const.^2 \sigma$$

(32)

(30a) becomes

$$\mathrm{Re}(\lambda_n) \approx \frac{\int_\Omega (\| \nabla A_n \|^2 + 1/2 \, const.^2 \| A_n \|^2) \sigma d\Omega}{\int_\Omega \| \partial A_n / \partial p \|^2 d\Omega} = \frac{I_n^{postive} - I_n^{negative}}{\int_\Omega \| \partial A_n / \partial p \|^2 d\Omega}$$

(33)

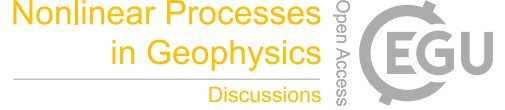



Here,

$$I_n^{postive} = \int\limits_{\Omega_{\sigma \geq 0}} (\| \nabla A_n \|^2 + 1/2\, const.^2 \| A_n \|^2)\sigma d\Omega \qquad (34a)$$

$$I_n^{negative} = \int\limits_{\Omega_{\sigma < 0}} (\| \nabla A_n \|^2 + 1/2\, const.^2 \| A_n \|^2)| \sigma | d\Omega \qquad (34b)$$

are integrations over two sub-domains with $\sigma \geq 0$ and $\sigma < 0$, respectively. As a result, if $I_n^{postive} \geq I_n^{negative}$,
then $\mathrm{Re}(\lambda_n) \geq 0$. Otherwise if $I_n^{postive} < I_n^{negative}$ we have $\mathrm{Re}(\lambda_n) < 0$. A particular case in which
approximate relation (32) becomes unnecessary is that $\sigma = \sigma(p)$ and $\nabla \sigma = \Delta \sigma = 0$, namely $\sigma$ is
horizontally uniform. From (30) we have

$$\mathrm{Re}(\lambda_n) = \frac{\int\limits_{\Omega} \sigma \| \nabla A_n \|^2 d\Omega}{\int\limits_{\Omega} \| \partial A_n / \partial p \|^2 d\Omega} = \frac{I_{postive} - I_{negative}}{\int\limits_{\Omega} \| \partial A_n / \partial p \|^2 d\Omega} \qquad (35a)$$

$$\mathrm{Im}(\lambda_n) = 0 \qquad (35b)$$
Then $\lambda_n$ is real, and in such situation

$$I_n^{postive} = \int\limits_{\Omega_{\sigma \geq 0}} \sigma \| \nabla A_n \|^2\, d\Omega \qquad (36a)$$

$$I_n^{negative} = \int\limits_{\Omega_{\sigma < 0}} | \sigma | \| \nabla A_n \|^2 )d\Omega \qquad (36b)$$

If $\sigma > 0$ everywhere, we have always $\mathrm{Re}(\lambda_n) > 0$. But if $\sigma < 0$ in some domain of the atmosphere, one can
always find both modes with $\mathrm{Re}(\lambda_n) > 0$ and modes with $\mathrm{Re}(\lambda_n) < 0$. We will show below these two

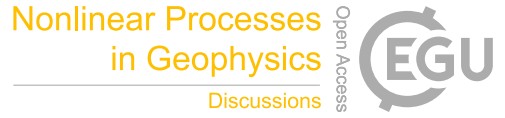

kinds of modes roughly belong to IGWs and free mode of convection, respectively. Equation (29) has
solutions in the form of $T_n = C_n e^{\Lambda_n t}$. By substituting the form into (29), we have

$$\Lambda_n = \pm \sqrt{[-\text{Re}(\lambda_n) - f^2]^2 + [-\text{Im}^2(\lambda_n)]}\, e^{i\frac{\theta_n}{2}} \qquad (37)$$

where

$$\theta_n = \begin{cases} \arctan[\dfrac{-\text{Im}(\lambda_n)}{-\text{Re}(\lambda_n) - f^2}]; & \text{Re}(\lambda_n) < -f^2 \ (Convection) \\[2ex] \arctan[\dfrac{-\text{Im}(\lambda_n)}{-\text{Re}(\lambda_n) - f^2}] + \pi; & \text{Im}(\lambda_n) > 0 \ and \ \text{Re}(\lambda_n) > -f^2 \ (IGW) \\[2ex] \arctan[\dfrac{-\text{Im}(\lambda_n)}{-\text{Re}(\lambda_n) - f^2}] - \pi; & \text{Im}(\lambda_n) < 0 \ and \ \text{Re}(\lambda_n) > -f^2 \ (IGW) \end{cases} \qquad (38)$$

There is a simple way to determine directly the property of a mode $\Lambda_n$. Since $\nabla \sigma$ is usually very small,
$\text{Im}(\lambda_n)$ is also small as compared with Re $(\lambda_n)$. So by letting $\text{Im}(\lambda_n) = 0$, unstable and stable mode given
by (29) are regarded as convection and IGW mode, respectively. The existence of $\text{Re}(\lambda_n)$
$< -f^2$ corresponds to the case of $\sigma < 0$ in some domain of the atmosphere and can be shown as unstable
modes describing free convection. Also, modes with $\text{Re}(\lambda_n) > -f^2$ belong to IGWs that coexisting with
modes of free convection. As $f^2$ is relatively a small parameter, it is usually negligible.
For modes with $\text{Re}(\lambda_n) < -f^2$, it can be inferred from the relation $I_n^{postive} < I_n^{negative}$ that strong
convective activities which can be measured by the average value of $\| \nabla A_n \|^2 + 1/2\, const.^2 \| A_n \|^2$, are
mainly confined to the finite domain of $\sigma < 0$. In fact, domain of stable stratification is usually greater
than that of unstable stratification, therefore $I_n^{postive} < I_n^{negative}$ gives





$$\frac{(\overline{\| \nabla A_n \|^2 + 1/2\, const.^2 \| A_n \|^2})_{\Omega_{\sigma<0}}}{(\overline{\| \nabla A_n \|^2 + 1/2\, const.^2 \| A_n \|^2})_{\Omega_{\sigma\geq0}}} > \frac{\Omega_{\sigma\geq0}}{\Omega_{\sigma<0}} \gg 1 \tag{39}$$

where overbar denotes the weighted average by $|\sigma|$ over corresponding domains. The physical meaning
can be explained in a simple way that convection modes are trapped by domain of $\Omega_{\sigma<0}$. Similarly, it
can be shown that modes of IGWs with $Re(\lambda_n) > 0$ coexisting with convection mode in this domain of
$\sigma < 0$ need not to be trapped and can radiate outward. A very interesting fact can be inferred from
$Re(\Lambda_n)$ that IGWs become slowly-growing unstable modes in addition to oscillating and propagating
as long as $Im(\lambda_n) \neq 0$ due to $\nabla\sigma \neq 0$. This means IGWs need not to be generated by convective activities,
but it is rather generated by the horizontal inhomogeneity of the stratification. For the same reason,
convection modes in this case become slowly oscillatory in addition to rapidly growing.
Finally, the inhomogeneous part of equation (26) can be investigated by projecting (26) onto $A_{npp}(x, y,$
$p)$, which gives
$$\frac{d^2 T_n}{dt^2} + (\lambda_n + f^2)T_n = R_n(t) \tag{40}$$
It describes both forced convection and the two-way interaction between convection and imbalance
forcing. Also further study on the triggering of convection can employ (40) as the starting equation.
*4.2 Generalization for symmetric inertial instability*
The results in section 4.1 can be generalized to the case of symmetric inertial instability given jointly
by the equation in table I and (24) as

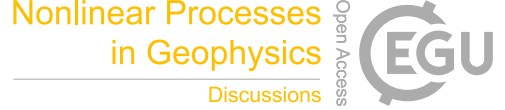

$$\frac{\partial^2 \bar{\delta}_{pp}}{\partial t^2} + \sigma \nabla^2 \bar{\delta} + f(f + \varsigma_0) \bar{\delta}_{pp} = \bar{\Re}(\varsigma, \varphi) \qquad (41)$$

4    Similarly, it is easy to show convection modes are defined by eigenvalue problem as below

$$\sigma \nabla^2 A_n = [\lambda_n - f(f + \varsigma_0)] A_{npp} \qquad (42)$$

8    Some procedures analogous to those in section 4.1 give

$$\mathrm{Re}(\lambda_n) = \frac{\lambda_n + \lambda_n^*}{2} = \frac{\int_\Omega [\sigma \|\nabla A_n\|^2 - 1/2 \|A_n\|^2 \Delta \sigma + f(f + \varsigma_0)\|\partial A_n/\partial p\|] d\Omega}{\int_\Omega \|\partial A_n / \partial p\|^2 d\Omega} \qquad (43a)$$

$$\mathrm{Im}(\lambda_n) = \frac{\lambda_n - \lambda_n^*}{2} = \frac{\int_\Omega \nabla \sigma (A_n^* \nabla A_n - A_n \nabla A_n^*) d\Omega}{2\int_\Omega \|\partial A_n / \partial p\|^2 d\Omega} \qquad (43b)$$

One can further discuss free and forcd convection mode of symmetric inertial instability based on this

result.

**5 Meso-scale convection with horizontally uniform $\sigma$**
The particular case of $\sigma$ being $p$-dependent only in (26) will be discussed in detail in this section, not
only because our previous results in this case need to be modified, but also because the applications
become possible in the analysis of meso-scale systems such as those in tropical region where a horizontally
uniform unstable $\sigma$ in lower atmosphere is a good approximation.



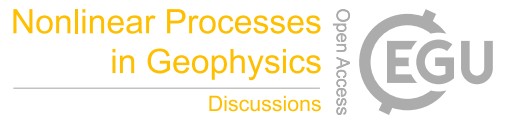

*5.1 Analytical solution: convection and IGWs*
By projecting (26) on $P_n$, the vertical modes defined by the eigen-system

$$-\frac{d^2 P_n}{dp^2} = \lambda_n \sigma P_n; \quad n = 0, 1, 2, \cdots \qquad (44)$$

satisfying suitable lower and upper boundary conditions (Zhao *et al.*, 2011), we obtain

$$\frac{\partial^2 \delta_n}{\partial t^2} - c_n^2 \nabla^2 \delta_n + f^2 \delta_n = \Re_n(\varsigma, \varphi) \qquad (45)$$

Here, $c_n^2 = 1/\lambda_n$. For stable vertical modes with $c_n^2 > 0$, (45) describes the IGWs and the spontaneous
emission (Lighthill,1952; Ford et al.,2000). Whereas for unstable vertical modes with $c_n^2 < 0$, (45)
describes convection. It may be quite common that for a stratification being unstable in some certain layer,
we have both eigenvalues of $c_n^2 < 0$ and $c_n^2 > 0$, convection and IGW modes coexist in one region. An
very interesting fact can be inferred from the last section is the vertical modes of convection are trapped in
the   layer of $\sigma < 0$, and so are modes of free and forced convection. So, actual convection comes out to be
the sum of both modes, namely

$$\delta = \sum_{c_n^2 > 0} \delta_n + \sum_{c_n^2 < 0} \delta_n \qquad (46)$$

A convection mode of $c_n^2 < 0$ is the super position of homogeneous and inhomogeneous solutions of (45),
corresponding to free and forced convection, respectively. It is given previously by Zhao *et al.* (2011) as

$$\delta_n(\mathbf{r}, t) = A_n \exp(k_x x + k_y y + \omega t)i - \frac{1}{4\pi c_n^2} \int_{t' < t} \int_{-\infty}^{\infty} \int_{-\infty}^{\infty} \Re_n(\mathbf{r}', t') \mathrm{G}(\mathbf{r} - \mathbf{r}', t - t') d\mathbf{r}' dt' \qquad (47)$$

The growth rate of the free unstable mode is obtained from the dispersion relation as
$\lambda = i\omega = \sqrt{-(k_x^2 + k_y^2)c_n^2 - f^2}$ , while forced convection are given by the Green's function

$$G(\mathbf{r},\mathbf{r}',t,t) = \frac{1}{4\pi} \frac{\exp[i\frac{f}{\sqrt{-c_n^2}}\sqrt{|\mathbf{r}-\mathbf{r}'|^2 - c_n^2(t-t')^2}]}{\sqrt{|\mathbf{r}-\mathbf{r}'|^2 - c_n^2(t-t')^2}} \qquad (48)$$

where $\mathbf{r} = x\mathbf{i} + y\mathbf{j}$, and the causality demands $t > t'$. Rather than just these portions were considered
previously, IGW modes of $c_n^2 > 0$ also need to be considered in this paper as the superposition of its free
IGW mode (homogeneous solution) and spontaneous emission of IGW (inhomogeneous solutions). Unlike
free modes of convection, free modes of IGW are stable and unable to grow, so we just need to consider the
spontaneous emission of IGW which is given by
$$\delta_n(\mathbf{r},t) = \frac{c_n^2}{4\pi} \int_{c_n(t-t')>|\mathbf{r}|} \iint_\Omega \Re(\mathbf{r}',t')G(\mathbf{r}-\mathbf{r}',t-t')d\mathbf{r}'dt' \qquad (49)$$

where the Green's function $G$ is (see Appendix C)
$$G(\mathbf{r}-\mathbf{r}',t-t') = \begin{cases} \dfrac{c_n^2}{4\pi} \dfrac{\exp[i\frac{f}{c_n}\sqrt{c_n^2(t-t')^2 - |\mathbf{r}-\mathbf{r}'|^2}]}{\sqrt{c_n^2(t-t')^2 - |\mathbf{r}-\mathbf{r}'|^2}}, & |\mathbf{r}-\mathbf{r}'| < c_n(t-t') \\[3em] \dfrac{c_n^2}{4\pi i} \dfrac{\exp[-\frac{f}{c_n}\sqrt{|\mathbf{r}-\mathbf{r}'|^2 - c_n^2(t-t')^2}]}{\sqrt{|\mathbf{r}-\mathbf{r}'|^2 - c_n^2(t-t')^2}}, & |\mathbf{r}-\mathbf{r}'| > c_n(t-t') \end{cases} \qquad (50)$$



Here $\Omega$ denotes the integrating domain $|\mathbf{r}-\mathbf{r}'|<c_n(t-t')$. The Green's function in domain
$|\mathbf{r}-\mathbf{r}'|>c_n(t-t')$ remains imaginary and has no contribution to the field of divergence. The physical
meaning is explained as the amplitude of IGW at time $t$ in place $\mathbf{r}$ is decided by the accumulation of the
forcing effect at all the earlier time $t'$ in everywhere $\mathbf{r}'$. If $|\mathbf{r}-\mathbf{r}'|=c_n(t-t')$ is viewed as the pure wave
front of IGW, (49) indicates that its influence is inversely proportional to the distance between $\mathbf{r}$ and the
wave front. Moreover, the influence of the wave front at $\mathbf{r}$ after it overtakes this point is mainly inertial
oscillation, while it has no influence before the overtaking. Overall, IGW behaves primarily as the
disturbance around the wave front.
*5.2 Deep structures of convective activities inside and outside meso-scale system*
The analytical inhomogeneous solutions of our studies seem too complicate in form to have practical
use. Nevertheless, they have approximate forms that are applicable for the straightforward estimation of
convection/IGW structures inside and outside a meso-scale system as follows.
Inside the meso-scale imbalanced vortical flow region, the Froude number $Fr$ for a meso-scale system
can be estimated by $Fr = U / \sqrt{gH}$. $U \sim 10^1 m/s$ and $H \sim 10^4 m$ are the wind scale and vertical scale,
respectively. So we have $Fr \ll 1$ and can demand $Fr \propto U/c_n \ll 1$ in (45), which confines the present
discussion to the deep structure of meso-scale convection system. We have also the Rossby number
$\mathrm{Re} = O(1)$, and let $L$ be the scale of the imbalance of meso-scale system, thus a comparison between the
magnitudes of different terms of the left side hand of (45) gives

$$\frac{\partial^2/\partial t^2}{c_n^2 \nabla^2} \propto \frac{1/T^2}{c_n^2/L^2} = \frac{U^2}{c_n^2} = F_r^2 \ll 1 \qquad (51a)$$

$$\frac{f^2}{c_n^2 \nabla^2} = \frac{f^2}{c_n^2/L^2} = \left(\frac{F_r}{\mathrm{Re}}\right)^2 \ll 1 \qquad (51b)$$

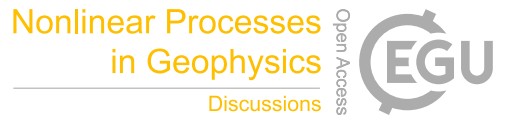

As a result, (45) becomes

$$-c_n^2 \nabla^2 \delta_n \approx \Re_n(\varsigma, \varphi) \qquad (52)$$

Since both forced convection modes with $c_n^2 < 0$ and IGW modes with $c_n^2 > 0$ contribute to the
convergence or the vertical motion, they should be viewed as the components of convection. The structure
of deep convection inside the meso-scale system can be estimated simply by

$$\delta_n \propto -\nabla^2 \delta_n \propto \begin{cases} \Re_n(\varsigma, \varphi)/c_n^2, & c_n^2 > 0 \\[2ex] -\Re_n(\varsigma, \varphi)/c_i^2, & c_n^2 = -c_i^2 < 0 \end{cases} \qquad (53)$$

Anyway, this structure just applies to forced convection. As an example to illustrate convective responses
that the (53) would predict, we consider an idealized system with a moving upper-level disturbance. A
moving upper-level disturbance can result in vorticity or temperature advection, which is an imbalance
forcing. After projected onto vertical modes, the forced modes of convection and IGW are estimated by the
structures of such imbalance forcing according to (53). Although forced IGW modes contribute also to
vertical motion, an essential difference is that forced convection modes are trapped in the unstable layer of
stratification. In addition, free modes of convection contribute to the spatial scales of the convective
activities as well. They tend to select the smallest scales to develop and are embedded in the forced
convection.
Outside the meso-scale system, forced convection/IGWs with scales much larger are always induced. As
was suggested by Ford (2000), the imbalance forcing is confined to a meso-scale region with diameter $L$. If
wind speed scale is $U$, then that of temporal variations is $L/U$. So growth rate of the free mode of



convection with wavenumber $k$ can be estimated by the dispersion relationship, i.e. $\lambda = \sqrt{c_i^2 k^2 - f^2}$, while
the frequency of the free mode of inertia-gravity waves can be estimated by $\omega = \sqrt{c_n^2 k^2 + f^2}$. We can
assume both of them are proportional to $L/U$, namely the time scale of the variation of the vortical flow as
the source of forcing. So the scale of forced convection as well as IGWs outside the meso-scale system is
$2\pi L / Fr >> L$ when $Fr << 1$. As suggested by the Green's function of forced convection and IGWs, these
structures may move toward and outward the source of forcing, respectively (Zhao et al 2010; 2011).
*5.3 Approximate horizontal structure outside a meso-scale system*
The form of Green function solution in section 5.1 can be transformed into an easier way to estimate the
structure outside a meso-scale system. The structure of imbalance forcing $\Re(\varsigma, \varphi)$ might be very complex.
In order to reflect the influence of such asymmetry of $\Re(\varsigma, \varphi)$ over the outside region of the meso-scale
system, it is convenient to introduce its multipole description such as monopole, dipole and quadrupole as

$$m(t') = \iint_D \Re_n(\mathbf{r}',t') dS \qquad (54a)$$

$$\mathbf{p}(t') = \iint_D \Re_n(\mathbf{r}',t') \mathbf{r}' dS \qquad (54b)$$

$$\bar{\mathbf{q}}(t') = \iint_D \Re_n(\mathbf{r}',t') \mathbf{r}' \mathbf{r}' dS \qquad (54c)$$

18    respectively. Here, $D$ is the area covered by system and *r'* is a point within it. They are all functions of time

19    *t'*. Suppose Green's function is $G(\mathbf{r}-\mathbf{r}', t-t')$, so the forced solution of (45) with *r* far from $D$ or

20    $\|\mathbf{r}'\|/\|\mathbf{r}\| \propto F_r << 1$ at any time *t* can be approximated by multipole expansion

$$\delta_n(\mathbf{r}, t) = \int_{t>t'} \iint_D \Re_n(\mathbf{r}',t') G(\mathbf{r}-\mathbf{r}', t-t') d\mathbf{r}' dt'$$
$$= \int_{t>t'} [m(t') G_0(\mathbf{r}, t-t') - \mathbf{r}\cdot\mathbf{p}(t') G_1(\mathbf{r}, t-t') + \mathbf{r}\mathbf{r}:\bar{\mathbf{q}}(t') G_2(\mathbf{r}, t-t') + ...] dt' \qquad (55)$$

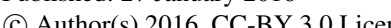


Here, $G_0$, $G_1$, $G_2$ and *etc.* are determined by $G$ and their forms are not given here for the simplicity. As a
result, the far-field response at $O(1)$, $O(F_r)$ and $O(F_r^2)$ are determined by $m(t')$, $\mathbf{p}(t')$ and $\tilde{\mathbf{q}}(t')$,
respectively. If strong imbalance comes out for some reason, this imbalance will cause far-field IGW and
convection $\delta_n(\mathbf{r},t)$ as nearly free wave, whose dispersion tends to weaken itself, i.e. $\delta_n(\mathbf{r},t) \to 0$ as $t$
$\to \infty$. The only way for that is $\Re(\varsigma,\varphi) \to 0$ as $t \to \infty$. So it tends to remove imbalance on the contrary.
This is known as the process of balanced flow adjustment. The compromise between these two tendencies
of generating and reducing of imbalance turns out to keep the imbalance and its radiation given mainly by
the quadrupole radiation (Ford, 2000). So, lower order monopole or even dipole imbalance will be
eliminated immediately by the process of balanced flow adjustment. This fact can be useful in the
understanding regarding some of typhoon's behaviors in the next section.
**6 Typhoon properties as an example of potential application**
For the purpose of application, we need to know the physical meaning of the imbalance forcing $\Re(\varsigma,\varphi)$
in (3). This imbalance forcing has something to do with the spatio-temporal inhomogeneities of the basic
flow, which are called systems by meteorologists. Thus it may be viewed as a forcing upon convection by
meso-scale system. The term $\Re(\varsigma,\varphi)$ is also closely related to the process of the adjustment of
geostrophic and gradient flows as was discussed in the end of the last section. If the vertical or horizontal
structures of the terms in brackets in $\Re(\varsigma,\varphi)$ are approximately sine or cosine functions, the imbalance
forcing can be roughly replaced by the terms of $(a_\varsigma^2 + b_\varsigma^2 - \varsigma^2)_t$ (nonsteady processes of the vortical flow),
$\mathbf{V}_\varsigma \cdot \nabla \varsigma$ (vorticity advection by the vortical part of the flow) and $[\mathbf{V}_\varsigma \cdot \nabla(\frac{\partial \varphi}{\partial p})]_p = -[\mathbf{V}_\varsigma \cdot \nabla \nu]_p$ (difference of
specific volume $\nu$ advections by the vortical flow between upper and lower levels). Notice that a constant
background wind field has been absorbed into the vortical part of the flow $\mathbf{V}_\varsigma$.

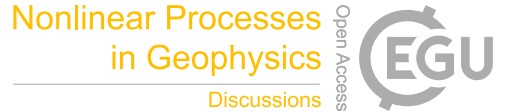

Usually, study on meso-scale convection stresses the role of water vapor and condensation. Emanuel et
al.(1994) argue that the overall effect of moist or diabatic convection on larger-scale circulations can be
viewed just as a reduction (by roughly an order of magnitude) to the effective static stability of such
circulations. Similarly, since our framework applies to meso-scale convective activity rather than small
scale cells as is mentioned in section 2, in the following case of typhoon, as a meso-$\alpha$-scale system, we can
assume a horizontally-averaged water vapor distribution $\overline{q}(p)$, which is steady in time but changes with
height. The condensation heating is assumed as $Q_c = -L\dfrac{\partial \overline{q}}{\partial p}\omega$. By substituting it into thermodynamic
equation (A1c) in Appendix A (i.e. $\dfrac{\partial}{\partial t}(\dfrac{\partial \varphi}{\partial p}) = -\sigma\omega - \mathbf{V}\cdot\nabla(\dfrac{\partial \varphi}{\partial p})$), we obtain an effective static stability
parameter $\sigma_e = \sigma - \dfrac{LR}{c_p p}\dfrac{\partial \overline{q}}{\partial p}$ . So we can introduce an effective static stability to reflect roughly the effect
of moist or diabatic convection, and above discussion seems unchanged in form. Therefore, the concept of
imbalance forcing of both convection and IGWs, or more accurately the interaction between imbalanced
vortical flow and convection/IGW modes, has potential applications in many aspects of meso-scale systems
with or without moisture.
As a simple example of application, this theory may be supportive in understanding some key issues of
typhoon study such as typhoon's organization and the relationship between typhoon recurvature (a sudden
turning of typhoon track, see *e.g.* Chen et al, 2002) and typhoon's asymmetric structure. The change of
meso-scale disturbances associated with typhoon is usually owed to linear barotropic instability or linear
inertial instability. See, *e.g.* Hendricks et al. (2009) for barotropic instability and Vigh and Schubert (2009)
for inertial instability. However, as was argued above, any mention of instability implies there is a
prescribed and balanced axisymmetric basic flow under the present circumstance, which excludes more
possibilities associated with the two-way interaction between convection/IGW and imbalanced basic state
of typhoon. In fact, it's difficult in real atmosphere to have a strict balanced axisymmetric solution ($\zeta_0$, $\varphi_0$)
as the basic state for a typhoon, due to its moving with shear flows, asymmetric latent and sensible heating,



the presence of surface friction, *β*-effect and *etc*. As a result, the imbalance forcing $\Re(\varsigma, \varphi)$ needs not to
vanish and its structure may be also asymmetric. So, the multipole description of far-field effects of
$\Re(\varsigma, \varphi)$ and the balanced flow adjustment theory of these multipoles stated in section 5.3 seem to be
useful for this purpose of further studies in application, such as issues of typhoon as below.
1) Typhoon's self-organization
Balanced flow adjustment theory of multipole description of imbalance can play an important role in the
process of typhoon's self-organization. First of all, an idealized mature typhoon tends to a balanced basic
flow which is steady in time and have a dynamically and thermo-dynamically axisymmetric structure. This
means there is no imbalance forcing and their multipoles due to $(a_\varsigma^2 + b_\varsigma^2 - \varsigma^2)_t = 0$, $\mathbf{V}_\varsigma \cdot \nabla \varsigma = 0$ as well as
$[\mathbf{V}_\varsigma \cdot \nabla v]_p = 0$ everywhere. However, for a small, but finite amplitude tropical disturbance, processes that
have been established to be essential elements in the self-organization of typhoon are moist convective
instability, vortical hot tower formation, vorticity aggregation and sea-to-air moisture exchange (see, e.g.,
Montgomery and Smith, 2014). Accompanied with these processes before a mature typhoon is finally
formed, the disturbance may have a strong imbalance due to the unsteady and asymmetric natures of
vortical flow of the disturbance. Such imbalance will produce forced modes of IGW/convection radiating
outwards and then give rise to balanced flow adjustment which diminishes the unsteady and asymmetric
imbalance natures of vortical flow and adjusts toward a steady and axisymmetric structure. So balanced
flow adjustment must be considered as another possible element in the self-organization of typhoon. Its key
role is to help other processes mentioned above attain an axisymmetric balance among them. This is even
more clearly seen and can be applied to the explanation of the self-organization process of typhoon's
structure as follows. If we write $\zeta$ (or $a_\varsigma$ and $b_\varsigma$) of the disturbance in polar coordinates $(r, \theta)$ with the
typhoon center being the origin of the coordinate system and expand in Fourier series
$$\varsigma(r, \theta, t) = \varsigma_0(r, t) + \varsigma_1(r, t)e^{i\theta} + \varsigma_2(r, t)e^{i2\theta} + \cdots \qquad (57)$$





These three terms are associated with the rotational monopole, dipole and quadrupole structures of $\zeta$ in
typhoon, respectively. However, they need not to be consistent with those of enstrophy $\zeta^2$, because it can be
shown that $\zeta^2$ has the flowing form
$$\varsigma^2 = \varsigma\varsigma^* = (\varsigma_0^2 + \varsigma_1^2 + \varsigma_2^2) + d(\varsigma_0,\varsigma_1,\varsigma_2)\cos[\theta - \theta_d(\varsigma_0,\varsigma_1,\varsigma_2)] + q(\varsigma_0,\varsigma_2)\cos 2[\theta - \theta_q(\varsigma_0,\varsigma_2)] + \cdots \quad (58)$$
Here, $d$ and $\theta_d$ describe the magnitude and phase of the dipole, respectively, while $q$ and $\theta_q$ the quadrupole.
These three parties in (58) are associated with monopole, dipole and quadrupole structures of enstrophy $\zeta^2$,
respectively. The self-organization process is thus related to the adjustment to remove the tendencies of
monopole, dipole and even quadrupole of enstrophy $\zeta^2$ (or $a_\varsigma^2 + b_\varsigma^2$ ). So we have roughly
$(\varsigma_0^2 + \varsigma_1^2 + \varsigma_2^2)_t \to 0$, $d(\varsigma_0,\varsigma_1,\varsigma_2)_t \to 0$ and $q(\varsigma_0,\varsigma_2)_t \to 0$, whose solution is $(\varsigma_0,\varsigma_1,\varsigma_2) \to const$. This
does no need to remove the monopole, rotational dipole and quadrupole of typhoon associated with $\zeta_0$, $\zeta_1$
and $\zeta_2$, although there may be enstrophy exchange among them. On the other hand, the balanced flow
adjustment tends also to remove the imbalance caused by $\mathbf{V}_\varsigma \cdot \nabla \varsigma \neq 0$. As mentioned in above, $\mathbf{V}_\varsigma \cdot \nabla \varsigma$
and its multipoles vanishes for an axisymmetric structure. So imbalance of $\mathbf{V}_\varsigma \cdot \nabla \varsigma \neq 0$ and its multipoles
are due to the asymmetric structure of $\zeta$ of the disturbance. The balanced flow adjustment then tends to
suppress the asymmetric portion such as $\zeta_1$ and $\zeta_2$ to develop further and allow axisymmetric structure $\zeta_0$
to grow up for a longer time. This is also the case for the axisymmetric thermodynamical structure
formation of height or temperature related to $[\mathbf{V}_\varsigma \cdot \nabla \nu]_p$. The corresponding observations stated by
Montgomery and Smith (2014) are as below. Since the process of spin-up is similar in almost all processes
known to be necessary for self-organization mentioned in above, involving the convectively-induced
inflow in the lower troposphere, which draws in absolute angular momentum surfaces to amplify the
tangential wind component. So the fact that these processes tend to select the axisymmetric component of

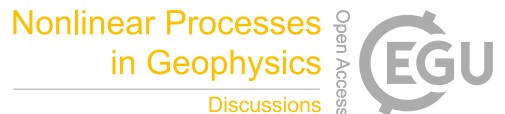

the disturbance to strengthen is attributed to the tendency of balanced flow adjustment to suppress
asymmetry. When $(\cdot)_t \to 0$ the finally-attained magnitude of $\varsigma_0$ is relatively larger than those of $\varsigma_1$, $\varsigma_2$
and *etc*., so a mature typhoon usually has a roughly axisymmetric basic structure and the dependence of $\varsigma_0$
on *r* determine the intensity and size of typhoon. The self-organization of typhoon and its convective
activity is then accomplished. Since balanced flow adjustment cannot reduce imbalance completely, a real
typhoon cannot be axisymmetric exactly, and dipole and quadrupole or even higher order multipole of
vorticity and deformation can exist as well, for which the existence of typhoon's spiral rainband may be
viewed as the evidence.
2) Mechanism of Fujiwhara effect
The Fujiwhara effect (Fujiwhara,1921) is following phenomena observed when two typhoons are in
proximity of one other. a) Their centers will begin orbiting cyclonically about a point between the two
typhoons. b)The two typhoons will be attracted to each other  and c) eventually spiral into the center point
and merge. It has not been agreed upon whether this is due to the divergent portion of the wind or vorticity
advection (DeMaria and Chan, 1984).
The mechanism of these phenomena can be again attributed to a process similar to that of typhoon's
self-organization in above. When two typhoons with almost equal size and intensity come close to each
other, they begin providing steering flow to each other and orbiting cyclonically about a point between the
two systems. If interaction between them are considered, they can be viewed as one single super typhoon
system with its center located at orbiting center between the two typhoons. The key point to consider such
issues is that the interaction between two nearly balanced typhoons can generate imbalance, while balanced
flow adjustment tends to select the way with a imbalance as weak as possible and will finally lead to a new
balance. Although analysis in analogy with that for self-organization in above can predict the result that
two typhoons may eventually merge into one single typhoon or phenomenon c), the details of process of



such adjustment are still necessary to explain phenomena mentioned previously. If orbiting center between
the two typhoons is chosen as the origin of the coordinate system, we can use (57) to describe the super
typhoon system. Notice $\zeta_l=0$ in this case, so the adjustment to remove the tendencies of monopole and
quadrupole of enstrophy $\zeta^2$ demands $(\varsigma_0^2+\varsigma_2^2)_t \to 0$ and $q(\varsigma_0,\varsigma_2)_t \to 0$, which determines a structure
$(\varsigma_0,\ \varsigma_2) \to const$. It doesn't need to mean one single typhoon can be formed before we are sure $\zeta_2$
diminishes essentially due to the adjustment to remove $\mathbf{V}_\varsigma \cdot \nabla \varsigma$, which will be demonstrated physically
as below by figure 1. Cyclonically orbiting of the two typhoons about the origin $o$ may cause positive
(negative) vorticity advection in front of (behind) each single typhoon with respect to the directions of their
motions. While in a Cartesian coordinate system, this distribution of vorticity advection has central
antisymmetric structure, i.e. the values of vorticity advection at $(x, y)$ and $(-x,-y)$ are the same, and
meanwhile it is also symmetric about two axes. So both monopole and dipole of vorticity advection vanish,
and primarily a quadrupole structure appears. Usually among monopole, dipole and quadrupole structures
of vorticity advection, quadrupole structure of vorticity advection produce relatively a weakest imbalance,
so the manner of interaction in a) which produce only quadrupole structure of vorticity advection is
selected as the way of interaction.. The magnitude of the quadrupole structure is proportional to both the
strength of vorticity advection regions and to the distance between a advection regions and the origin. The
former depends highly on the intensity of typhoons, while the latter the distance between two centers of
typhoons. Suppose the intensity of two typhoons remains almost unchanged, the balanced flow adjustment
tends to decrease the magnitude of the quadrupole of vorticity advection by reducing the distance between
two centers of typhoons, until it becomes zero and one single typhoon forms. This gives reasonable
explanations to phenomenon b) and c).
3) Asymmetry of typhoon and its motion
Imbalance forcing is also associated with vorticity advection by the vortical flow, namely, $\mathbf{V}_\varsigma \cdot \nabla \varsigma$. Here,



a spatially uniform but time-dependent background wind field has been absorbed into $\mathbf{V}_\varsigma$, which can be
selected as the wind speed of the center of typhoon and denoted by $\mathbf{V}_0(t)$. So

$$\mathbf{V}_\varsigma = \mathbf{V}_0(t) + \mathbf{V}'_\varsigma \qquad (59)$$

here $\zeta = \nabla \times \mathbf{V}'_\varsigma$. Thus, the monopole of the vorticity advection is given by

$$m(t') = \iint_D \mathbf{V}_\varsigma \cdot \nabla \varsigma \, dS \propto \mathbf{V}_0(t) \cdot \overline{\nabla \varsigma} + \overline{\mathbf{V}'_\varsigma \cdot \nabla \varsigma} = \left\| \mathbf{V}_0(t) \right\| \left\| \overline{\nabla \varsigma} \right\| \cos \alpha \qquad (60)$$

Hereafter, "$\overline{\phantom{x}}$" denotes $(1/D) \iint_D (\cdot) dS$. And $\alpha$ is the angle between the direction of typhoon movement
and the averaged vorticity gradient $\overline{\nabla \varsigma}$. Here, notice also $\overline{\mathbf{V}'_\varsigma \cdot \nabla \varsigma} = \overline{\nabla \cdot (\mathbf{V}'_\varsigma \varsigma) - \varsigma \nabla \cdot \mathbf{V}'_\varsigma} = \overline{\nabla \cdot (\mathbf{V}'_\varsigma \varsigma)} = 0$, if we
assume there is no net flux $\varsigma \mathbf{V}'_\varsigma$ into the area $D$ covered by typhoon through it edge. Suppose $\|\mathbf{V}_0(t)\|$ and
$\left\| \overline{\nabla \varsigma} \right\|$ are nearly constant, we have largest values of imbalance where $\alpha = 0$ or $\pi$, and smallest value of 0
where $\alpha = \pi/2$ or $-\pi/2$. Notice also that we have $\overline{\nabla \varsigma} = 0$ for an exactly axisymmetric typhoon. If the
structure of typhoon becomes asymmetric *i.e.* $\overline{\nabla \varsigma} \neq 0$ for some reason and there is a value $\alpha$ far enough
from $\pm \pi/2$ (quite often near to 0 or $\pi$ in observations, as will be shown subsequently), then strong
monopole of imbalance caused by vorticity advection comes out. According our previous discussions, the
balanced flow adjustment tends to eliminate this monopole part of imbalance immediately by adjusting $\alpha$
toward the nearer one of $\alpha = \pi/2$ and $-\pi/2$. Usually this process of balanced flow adjustment may be
accompanied with the turning of $\mathbf{V}_0(t)$ or typhoon recurvature. Such issue is also proposed as the effect of
interaction of typhoon with a steering flow upon typhoon track. If one decomposes $\overline{\nabla \varsigma}$ into $\overline{\nabla \varsigma_s}$ a
portion from steering flow, and $\overline{\nabla \varsigma_T}$ a portion from typhoon. It is easy to see that we have both $\overline{\nabla \varsigma_s} = 0$
and $\overline{\nabla \varsigma_T} = 0$ for an axisymmetric typhoon in a uniform steering flow, so typhoon will just move with
steering flow and no change of typhoon track can happen. Furthermore, if one assumes typhoon as an

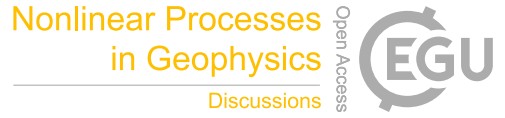

axisymmetric component of the stream, then $\overline{\nabla \varsigma_T} = 0$, and averaged vorticity gradient is mainly from
steering flow. In the latter case, previous research (see, *e.g.* Kasahara and Platzman, 1963; Demaria 1985)
find that vorticity gradient of steering flow results in a component of motion *π/2* to the left of the gradient,
together with a component in the direction of the gradient. On the contrary, in the case of weak steering
flow, which means both uniform component $\overline{U_s}$ and $\overline{\nabla \varsigma_s}$ of shear component are negligible for it, and
$\overline{\nabla \varsigma} \approx \overline{\nabla \varsigma_T}$, it is summed up from observations that there are four categories of asymmetric structure of the
typhoon that may be connected to typhoon recurvature in Chen et al. (2002). Upon their figures of observed
stream fields indicated in figure 2, we mark the averaged vorticity gradient $\overline{\nabla \varsigma}$ and the observed turning of
typhoon track by arrows in blue dashed line and red solid line, respectively. Our theory can give quite
reasonable explanation to this connection in all above cases. If we regard typhoons in figure 2 are
axisymmetric and asymmetries are from a weak steering flow with $\overline{U_s} \approx 0$ and $\overline{\nabla \varsigma_s} \neq 0$, we have
$\overline{\nabla \varsigma} \approx \overline{\nabla \varsigma_s}$ and conclusions remain the same, while turning of typhoon track is attributed to steering flow,
which can be another point of view. The results in Demaria (1985) may have different mechanism, because
a non-divergent barotropic model is employed and *β*-effect is considered.
Similarly, thermo-dynamical asymmetric structure of typhoon which may be caused by its dynamics and
by latent heat release as well as the distribution of low-boundary heating such as SST(sea surface
temperature), may also affect typhoon track. Because the difference of temperature or specific volume *v*
advections by the vortical flow between upper and lower levels gives also imbalance forcing, it can be
shown that

$$m(t') = \iint_D [\mathbf{V}_\varsigma \cdot \nabla v]_p \, dS \propto -[\|\mathbf{V}_0\| \|\overline{\nabla v}\| \cos \beta]_p \qquad (61)$$

In the derivation, $\overline{\mathbf{V}_\varsigma' \cdot \nabla v} = \overline{\nabla \cdot (\mathbf{V}_\varsigma' v)} - \overline{v \nabla \cdot \mathbf{V}_\varsigma'} = \overline{\nabla \cdot (\mathbf{V}_\varsigma' v)} = 0$ because we assume again there is no net flux
$v\mathbf{V}_\varsigma'$ into $D$ through it edge. $\beta$ is the angle between the moving direction of typhoon and $\overline{\nabla v}$ the

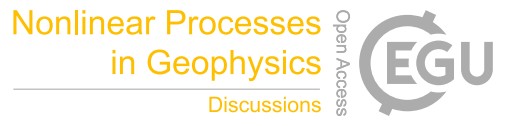

averaged specific volume gradient. If heating of upper atmosphere is weak, the structure of $v$ is
approximately axisymmetric and $\overline{\nabla v} = 0$, then we just need to consider the value of $\|\mathbf{V}_0\|\|\overline{\nabla v}\|\cos\beta$ at
lower atmosphere. So if typhoon track intersects the contour lines of $v$ or temperature, then there is
imbalance forcing. Again, the balanced flow adjustment tends to adjust $\beta$ toward $\pi/2$ or $-\pi/2$ so as to
weaken this imbalance. As a result, this process is accompanied with the tendency of the typhoon track
change. This can probably be used to explain how SST distribution affect typhoon track. Many researchers
stated in Chen et al. (2002) notice that a typhoon passing close to a warm sea region experiences some
deflection toward the warm sea region. Other researchers such as Jing (1996) also observed that typhoon
tends to move along the SST contour lines of the outside edge of a warm sea region. There seems so far no
good way to incorporate these two phenomena into giving a self-consistent explanation. We suppose these
facts can be simply explained as below. As is shown in figure 3, the red solid line with arrow represents the
track of a typhoon approaching a warm sea region. The distribution of SST around a warm sea region may
cause an averaged gradient of latent heating in the lower atmosphere due to evaporation and sensible
heating ones pointing to warm side of SST.
Typhoon may be sustained by latent heating, so the stronger latent heating in the warmer side enhances
warmer side of typhoon. As a result, a tendency appears to pull typhoon toward the warmer side of SST.
Meanwhile, it also implies that warmer side of typhoon has higher temperature and humidity in lower
atmosphere, and $\overline{\nabla v} \propto \nabla SST$ is a good approximation. In the early stage to approach warm SST region,
typhoon track intersects the contour lines (dark thin line) of SST at a large angle and induces an imbalance
to the vortical flow of typhoon. Consequently, balanced flow adjustment tends to get rid of this imbalance
by compelling typhoon track to follow the SST contour lines around a warm sea region.
The averaged vorticity gradient and temperature gradient can change typhoon track jointly, although we
discuss them independently so far. Then a question arises: which one is more important or effective to the
change of typhoon track? Since vorticity gradient is usually caused by dynamical process and varies more





rapidly than temperature gradient that is caused mainly by thermo-dynamical process associated with
heating, we suppose vorticity gradient may be relatively important or effective in altering the moving
direction of typhoon. Another issue is that typhoon may have both apparent instability process and
adjustment process associated with imbalance of basic flow. However, as discussed in section 3, at least
linear apparent instability cannot react on the imbalanced basic flow ($\zeta$, $\varphi$) defined in this study and is thus
almost not involved in adjustment process. It is also necessary to point out that one should be careful in
explaining the dynamics of vortex motion on a $\beta$-plane by above theory, because the theory so far is still
limited to $f$-plane.
**7 Summary and conclusions**

11       In the study of the convective activities of a meso-scale system, there is a big contradiction between the

aspiration to apply classic theory of instabilities and the unsatisfactory conditions of these theories due to
highly imbalanced natures of the basic states. By introducing an appropriately-defined imbalanced vortical
flow as the basic state, our previous study in Zhao, et al (2011) has extended instability theories from
balanced to imbalanced flows. This revised theory of instability considers not only the apparent instability
of the imbalanced basic state but also the two-way interaction between the convective activities and this
imbalanced basic state. In the linear sense, it is argued that convection can be regarded as the superposition
of free modes of convection and the response to the forcing induced by the imbalance of the basic flow.
Although the free modes of convection cannot act on the basic state, the forced part of convection do have
a reaction on the basic state though balanced flow adjustment and prevent it's imbalance from further
increase. Within the framework above, this paper makes progress in following key issues that we failed to
cope with previously.

23       Firstly, we have a further insight into the triggering mechanism of convection. A study by regular

perturbation method on the nonlinear case is performed for that purpose. It can be concluded that

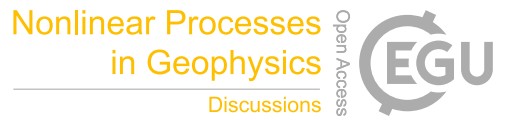

convection can be triggered in resonance either with imbalance forcing of basic state or with nonlinear
interaction among different modes of IGWs. Even if all these cannot happen, an imbalance forcing with
strong enough magnitude may eventually trigger convection after a time delay. These are essentially
different from the concept of Liyapunov instability in which an initial disturbance is necessary. So,
convection is more appropriately to be regarded as a spontaneous phenomenon without external initial
disturbance.

7        Secondly, in some simplified but relatively general dynamical setting for meso-scale system, the

influences of the inhomogeneity of stratification on convection and IGWs are explored. Qualitative
properties of free and forced modes of convection/IGW are investigated via an eigenvalue problem for
arbitrarily distributed stratification. It is found that modes of convection are trapped by domain of $\sigma < 0$, or
unstable stratification, while modes of IGWs coexisting with convection mode in this domain need not to
be trapped and can propagate through or radiate outward. *And due to horizontal inhomogeneity of the*
*stratification*, IGWs become slowly-growing unstable modes, in addition to oscillating and propagating,
and convection modes in this case become slowly oscillatory, in addition to rapidly growing. The specific
situation of horizontally uniform stratification which we considered previously in Zhao, et al. (2011) is
improved. Rather than just giving forced convection mode, all vertical motions that may contribute to the
structure of convection in meso-scale system are discussed, including those caused by free modes of
instabilities, forced convection and IGWs. The approximate forms that are applicable for the
straightforward estimation of convection/IGW structures inside and outside a meso-scale system are
derived. Particularly, a multi-pole description such as monopole, dipole and quadrupole is introduced to
link the induced far-field convection/IGW structures and the balanced flow adjustment inside a meso-scale
system, which is useful in our subsequent study on typhoon.

23       Finally, as a simple example to demonstrate the potential application of our theory on interaction

between convection/IGWs and its imbalanced basic state, the multi-pole description of imbalance forcing

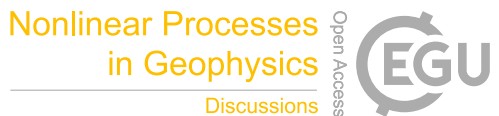

and then balanced flow adjustment, the central idea of our theory, gives reasonable explanations to key
issues in typhoon study such as the role of balanced flow adjustment in typhoon's self-organization,
Fujiwhara effect and the influence of typhoon's asymmetric structure on its track of motion.

**Acknowledgments**

This research was supported by the National Natural Science Foundation of China under Grant
41175065/40940022. We thank also Prof. Xiangde Xu at CAMS for his helpful discussion with the first
author on issues concerning typhoon track.

**Appendix A**

The basic equation are the vorticity equation, divergence equation and thermodynamic equation in
$p$-coordinates as below

$$\frac{\partial \varsigma}{\partial t} = -f\delta - \mathbf{V}\cdot\nabla\varsigma - \omega\frac{\partial\varsigma}{\partial p} - \varsigma\delta + \mathbf{k}\cdot(\frac{\partial\mathbf{V}}{\partial p}\times\nabla\omega) \tag{A1a}$$

$$\frac{\partial \delta}{\partial t} = f\varsigma - \nabla^2\varphi - \mathbf{V}\cdot\nabla\delta - \omega\frac{\partial\delta}{\partial p} - \frac{1}{2}(\delta^2 + a^2 + b^2 - \varsigma^2) - \frac{\partial\mathbf{V}}{\partial p}\cdot\nabla\omega \tag{A1b}$$

$$\frac{\partial}{\partial t}(\frac{\partial\varphi}{\partial p}) = -\sigma\omega - \mathbf{V}\cdot\nabla(\frac{\partial\varphi}{\partial p}) \tag{A1c}$$

17 Variables here have the same meanings with those in Section 2. From the continuity equation, vertical

18 velocity is related to the divergence $\delta$ by $\omega = \int_0^p \delta\,dp$.

**Appendix B**

20    Multiplying (28) by $A_n^*$, the complex conjugate function of $A_n$, we have

$$A_n^*\sigma\nabla^2 A_n - \lambda_n A_n^*\frac{\partial^2 A_n}{\partial p^2} = 0 \tag{B1}$$

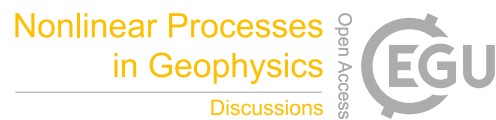



Also, multiplying the complex conjugate of (28), i.e.

$$\sigma \nabla^2 A_n^* - \lambda_n^* \frac{\partial^2 A_n^*}{\partial p^2} = 0$$

(B2)

by $A_n$ yields

$$A_n \sigma \nabla^2 A_n^* - \lambda_n^* A_n \frac{\partial^2 A_n^*}{\partial p^2} = 0$$

(B3)

Notice also that

$$\lambda_n A_n^* \frac{\partial^2 A_n}{\partial p^2} = \lambda_n [\frac{\partial}{\partial p}(A_n^* \frac{\partial A_n}{\partial p}) - \|\frac{\partial A_n}{\partial p}\|^2]$$ (B4a)

$$\lambda_n^* A_n \frac{\partial^2 A_n^*}{\partial p^2} = \lambda_n^* [\frac{\partial}{\partial p}(A_n \frac{\partial A_n^*}{\partial p}) - \|\frac{\partial A_n}{\partial p}\|^2]$$ (B4b)

and

$$A^* \sigma \nabla^2 A = \nabla \cdot (A^* \sigma \nabla A) - \sigma \|\nabla A\|^2 - A^* \nabla \sigma \cdot \nabla A$$ (B5a)

$$A \sigma \nabla^2 A^* = \nabla \cdot (A \sigma \nabla A^*) - \sigma \|\nabla A\|^2 - A \nabla \sigma \cdot \nabla A^*$$ (B5b)

Integrating (A1.1)+(A1.3) and (A1.1)-(A1.3) over domain $\Omega$, we have (30a) and (30b), respectively. Here,

$$\int_\Omega (\cdot) d\Omega = \iint_S \int_0^{p_s} (\cdot) dp dS; \quad \Omega = S \times [0, p_s]$$

(B6)

$S$ is the area of meso-scale system, while $p_s$ is surface pressure. In the derivation, the followingrelations are
employed

$$A_n^* \sigma \nabla^2 A_n - A_n \sigma \nabla^2 A_n^* = \nabla \cdot (A_n^* \sigma \nabla A_n - A_n \sigma \nabla A_n^*) + \nabla \sigma \cdot (A_n \nabla A_n^* - A_n^* \nabla A_n) \qquad (B7a)$$

$$A_n^* \sigma \nabla^2 A_n + A_n \sigma \nabla^2 A_n^* = \nabla \cdot (A_n^* \sigma \nabla A_n + A_n \sigma \nabla A_n^*) - 2\sigma \| \nabla A_n \|^2 - \nabla \sigma \cdot \nabla \| A_n \|^2$$
$$= \nabla \cdot (A_n^* \sigma \nabla A_n + A_n \sigma \nabla A_n^* - \| A_n \|^2 \nabla \sigma) - 2\sigma \| \nabla A_n \|^2 + \| A_n \|^2 \Delta \sigma \qquad (B7b)$$

And boundary conditions as below are considered as well

$$A_n^* \frac{\partial A_n}{\partial p}\bigg|_{p=0,\, p_s} = 0; \quad A_n \frac{\partial A_n^*}{\partial p}\bigg|_{p=0,\, p_s} = 0 \qquad (B8)$$

$$(A^* \sigma \nabla A + A \sigma \nabla A^* - \| A \|^2 \nabla \sigma) \cdot \hat{n}\big|_{s \to \infty} = 0 \qquad (B9a)$$
$$\sigma (A \nabla A^* - A^* \nabla A) \cdot \hat{n}\big|_{s \to \infty} = 0 \qquad (B9b)$$

## 14   **Appendix C**

16   Similarly to the way in Zhao and Gan (2010) for a barotropic model, the Green's functiom $G$ in (50) is

obtained as below. Since $G$ satisfies

$$\frac{\partial^2 G}{\partial t^2} - c_n^2 \nabla^2 G + f^2 G = \delta(\mathbf{r} - \mathbf{r}') \delta(t - t') \qquad (C1)$$

Its Fourier transform with respect to $t$ gives

$$\nabla^2 \tilde{G} + \frac{\omega^2 - f^2}{c_n^2} \tilde{G} = -\delta(\mathbf{r} - \mathbf{r}') e^{-i\omega t'} \qquad (C2)$$





Here $\tilde{G}$ is the Fourier transform of $G$. From the Green function of Helmholtz equation, one obtains

$$\tilde{G}(\mathbf{r}, \mathbf{r}', \omega) = \frac{i}{4} H_0^{(1)}(\sqrt{\frac{\omega^2 - f^2}{c_n^2}} |\mathbf{r} - \mathbf{r}'|) e^{-i\omega t'} \tag{C3}$$

6    Here $H_0^{(1)}$ is Hankel function. The inverse Fourier transform of (A2.3) gives the Green's function $G$ in (50),

7    in which following formulas are utilized in its derivation

$$F^{-1}[i\pi H_0^{(1)}(a\sqrt{b^2 - \omega^2})] = \frac{\exp(ib\sqrt{t^2 + a^2})}{\sqrt{t^2 + a^2}} \tag{C4a}$$

$$F^{-1}[\tilde{A}(\omega)e^{-i\omega\beta}] = A(t - \beta) \tag{C4b}$$

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

| Type of instability | Balanced basic flow | Equation describing instability | Remarks |
|---|---|---|---|
| Static instability | Static state | $$\frac{\partial^2 \delta_{pp}}{\partial t^2} + \sigma \nabla^2 \delta + f^2 \delta_{pp} = 0$$ | $\sigma < 0$ |
| Symmetric instability | Geostrophic flow $U$ ($x$-oriented) | $$\frac{\partial^2 \delta_{pp}}{\partial t^2} + N^2 \delta_{yy} - 2S^2 \delta_{yp} + F^2 \delta_{pp} = 0$$ | $N^2 = \sigma$, $S^2 = fU_p$, $F^2 = f(f + U_y)$ |
| Kelvin-Helmholtz instability | Parallel flow $U$ ($x$-oriented) | $$\frac{\partial^2 \delta_{pp}}{\partial t^2} + \sigma \nabla^2 \delta + f^2 \delta_{pp} + U \frac{\partial \delta_{xpp}}{\partial t} + U_p \frac{\partial \delta_{xp}}{\partial t} - fU_p \delta_{yp} = 0$$ | $U_y = 0$; $U_p \neq 0$ |
| Inertia instability | Gradient wind $U$($r$-oriented) | $$\frac{\partial^2 \delta_{pp}}{\partial t^2} + \sigma \nabla^2 \delta + f(f + \varsigma_0) \delta_{pp} + \varsigma_0 \frac{\partial}{\partial t} \frac{\partial \delta_{pp}}{\partial \theta} = 0$$ | $U_\theta = \varsigma_0 r$ $U_p = 0$ |

13    Table I. Types of linear instabilities, corresponding balanced basic flows and equations describing these

14    instabilities.


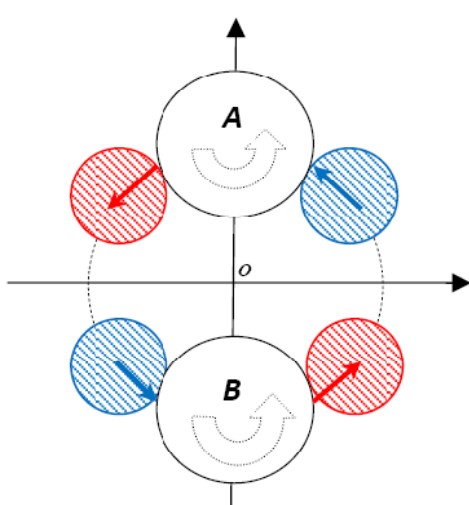

**Figure 1.** Cyclonically orbiting of the two typhoons *A* and *B* about the origin *o* may cause positive

(negative) vorticity advection in front of (behind) each single typhoon, with respect to the directions of

their motions. This distribution of vorticity advection has central antisymmetric structure, i.e. the values of

vorticity advection at ($x$, $y$) and ($-x$,$-y$) are the same, and meanwhile it is also symmetric about two axes. So

vorticity advection appears to be primarily a quadrupole structure.





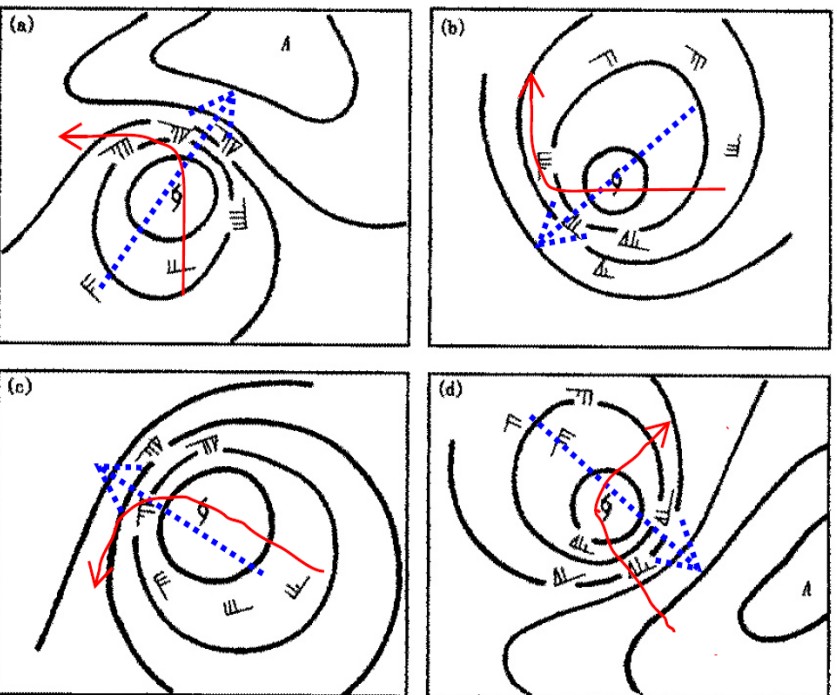


**Figure 2.** The averaged vorticity gradient $\overline{\nabla \varsigma}$ and the turning of typhoon track are marked by arrows in
blue dashed line and red solid line respectively upon the figure of stream fields in Chen et al. (2002 ).






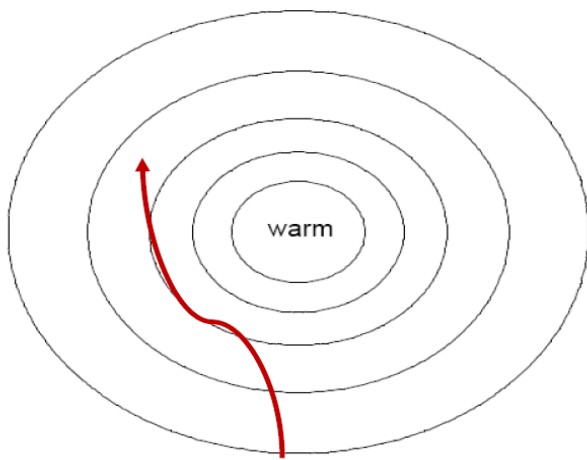

**Figure 3.** The arrow in red thick solid line represents a typhoon track approaching a warm sea region
indicated by the contour lines (dark thin line) of SST.

