# Peer review of "A possible theory for the interaction between convective activities and vortical flows"

_Nonlinear Processes in Geophysics, 2016_

## Referee Comment (RC1) · Anonymous Referee #1 · 26 Feb 2016

I writing to give you my impression about this paper and reasons why I am declining to provide any further comments nor a detailed report as suggested for this work.

I think the paper is poorly written and most of its statements are wrongly and poorly motivated. While the title and the core abstract suggest that this work is about meso- and synoptic-scale convection, there is nothing in its content that speaks about this subject. I don't see how someone can claim on studying atmospheric convection without involving moisture and precipitation or even some kind of thermal forcing such as radiation and/or surface heat and/or moisture fluxes. At best this work is about stratified turbulence and/or nonlinear interactions between gravity waves and slowly evolving vortical flows. These two subjects have been extensively studied during the last few decades

and the present work is far from making any new contribution of some kind. The mathematical study which is based on asymptotic expansions and looks at possible resonant interactions between gravity waves and the vortical motion; this is standard in this business and the authors have nothing new to offer. Moreover, I have serious doubts that the present work is of any use. The equation they use to built their theory, I quote, "is not closed". I don't see how someone can claim growing or decaying and balanced or imbalanced solutions for a non closed equation. Furthermore, the paper is poorly written and full of typos. For all these reasons, it must be rejected and I am reluctant to waste my time to write a detailed report to send to the authors or post online because it will counterproductive.

---

## Referee Comment (RC2) · Anonymous Referee #2 · 28 Feb 2016

**General Comments**

This submission is a follow up to Zhao et al (2011) and rests on Eq. 1 here, which was obtained in that earlier study. The equation is somewhat complicated but nonetheless amenable to analysis of its expected behaviours. That is what is attempted here for most of the submission. Then in the final section, there is an attempt to apply this thinking to certain aspects of typhoons.

The analysis is not always well explained or presented, and could probably be shortened by tightening it up. For example, as detailed below, although the various claims do appear quite reasonable, some things are not fully justified and are stated too strongly, with plausible assumptions being described almost as though they were actual proofs

of properties of the equation. Much of the analysis is nonetheless acceptable, although not particularly enlightening in isolation. I found myself awaiting the application rather impatiently in order to see whether working through the analysis would prove worthwhile. Alas, this was not the case.

The issue is that the authors offer no more than possible explanations of how a full set of calculations might be able to provide some explanation for properties of typhoons. Again, statements are overclaimed. Often this to the effect that things have ben explained when in fact the "explanation" has gone no further than speculations as to what aspects of the equation might be able to play what role in providing explanations.

Really the study is crying out for actual calculations to be done using the equations presented. I don't see that this would be an impractical task. Idealized simulations of typhoons could be used to provide suitable background data with which to solve the equations, and to test the authors' ideas and hypotheses properly. If that were to be done, there could ultimately be a strong paper that emerged, but it would very likely be a very different paper from the article submitted here.

**Specific Comments**

1. p5, line 14. The equations are stated in the Appendix but a summary of their properties is not given. It would have been useful to do so.

2. p7, line 17, "it is proven" does not seem to be the right wording here. Rather the statement follows simply from what the authors mean by balanced or imbalanced in this context.

3. Eq. 1. Much of the analysis in the paper ignores the nonlinear operator on $\delta$, specifically $\mathcal{I}(\delta)$ in Eq. 2b. However, this needs some more motivation and discussion in order to clarify the circumstances under which this will or will not be a reasonable assumption. On page 9, line 6 for instance we are told that this is done "for simplicity" and there is a little discussion at the end of the same section.

However, the assumption seems to need more attention and should be stated much more clearly and strongly, with the caveats noted in the Introduction and Conclusions.

4. The greek letter $\zeta$ is not used consistently: see for example the different forms used on page 9, lines 20 and 21. This is not unusual and the authors need to go through the complete text to check.

5. Eqs. 14 and 15. The quantities $\mathcal{I}_2$ and $\mathcal{I}_i$ need to be properly defined.

6. Eq. 32. This assumption is introduced without comment. It is a reasonable approach to take in the analysis but it does need to be properly introduced, motivated and discussed, perhaps at the start of the Section.

7. p19, lines 18-19. At this stage it is far from clear that this statement about negative $\sigma$ should hold true. Only later do we learn about the authors' arguments for it, and further that this statement is not necessarily true but simply a plasuible assumption. The argument is made at the end of page 20, and I have no complaint about it as a plausible assumption. But again it should not be stated as something stronger. It is unlikely but not out of the question that the area of negative $\sigma$ may be rather large, or that the modulus of the $A_n$ variations could be rather small. So the "inferred" for example, is not appropriate.

8. p20, line 15. Small relative to what, and with what justification?

9. Eqs. 49 and 50. A comparison of these with Eq. 47 suggests an error sonewhere here, given that $c_n$ is a dimensional quantity!

10. p29. The treatment of condensational heating needs much more discussion and motivation when it is introduced. Moreover it needs proper specification. Ascent is always staurated, but what about descent. The relation stated seems to imply latent cooling during descent which would be a strange assumption.

**Technical/Minor Corrections**

1. p4, line 16, do not.

2. p7, line 19, remains should be singular.

3. p8, line 24, this should be reworded.

4. Eq. 7, it would be helpful to clarify here that the asterisk indicates an adjoint. In the current text, this only becomes clear a few pages later.

5. p13, line 14. Reword this sentence.

6. p14, line 11. quadratic.

7. p15, line 12, known should read shown.

8. Eq. 36b. the brackets and modulus signs need fixing.

9. p22, line 13, forced.

10. Some of the equation referencing seems to have gone awry:

    (a) p21, line 23. Eq. (24) is not right. 26?
    (b) p40, line 18.
    (c) p42, line 6

11. Eq. 48. The final $t$ in the argument list for $G$ should read $t'$

12. p25, line 10, complicated.

13. p25, line 17, and elsewhere. To avoid the obvious potential for confusion the Rossby number should be denoted by the standard Ro and not by Re.

Interactive
comment
14. p27, line 20 and elsewhere. $F_r$ is a poor choice of notation, since there could be scope for confusion with a Froude number.

15. p29, line 22, it is.

16. p31, line 9, parties should read parts.

17. p33, line 5, does not.

18. p35, line 16. Reword, the SST is not a heating.

19. p37, line 20, it is.

20. Eqs. B5, the $n$ subscripts are missing.

---

## Author Comment (AC1) · 7 Mar 2016

Reply to Referee #1

I writing to give you my impression about this paper and reasons why I am declining to provide any further comments nor a detailed report as suggested for this work. I think the paper is poorly written and most of its statements are wrongly and poorly motivated.

Reply: Sorry for my late reply. In fact, I'm wondering why you can make such totally negative comments on our work. I find your negative comments are mainly about the basic ideas following up our previous NPG paper (i.e. Zhao et al, 2011). These issues

were fully discussed with two reviewers and editor of Zhao et al (2011). I think if the editor can still find their comments of that time, he will see very different comments from yours. I provided a copy of Zhao et al (2011) when I submitted the present paper, because this work is a continue of Zhao et al (2011). However, I'm not sure if you have received or read it. Even so, I'd like to explain further as below.

While the title and the core abstract suggest that this work is about meso- and synoptic-scale convection, there is nothing in its content that speaks about this subject. I don't see how someone can claim on studying atmospheric convection without involving moisture and precipitation or even some kind of thermal forcing such as radiation and/or surface heat and/or moisture fluxes.

Reply: A fully moisture convection is too difficult issue and have never been understood well even by people of numerical model. But this does not need to mean we cannot use dry convection or some highly simplified moisture convection in the theoretical study of the dynamical aspects of meso-scale system (in the typhoon study of our paper we do introduced a simplified moisture heating). Otherwise, all early theoretical studies such as symmetric instability become meaningless. This is obviously not true. Are you sure theoretical studies on symmetric instability are nothing? If not, please check whether most of them consider moisture convection exactly or not.

At best this work is about stratified turbulence and/or nonlinear interactions between gravity waves and slowly evolving vortical flows. These two subjects have been extensively studied during the last few decades and the present work is far from making any new contribution of some kind.

Reply: I think this is the key point why our work is not understood properly. There are two kind of modes in the stratified atmosphere with unstable domain of stratification or other instability. Basically, unstable modes are regarded as convection modes, while stable modes the inertial-gravity waves. The work is mainly about interaction between (at least dry) convection modes (turbulence is at too small scale) and the slowly evolving vortical flows. It also deal with interaction between inertial-gravity waves and the slowly evolving vortical flows. Our new contribution are that : in the former case of convection modes, we performed a study on the triggering mechanism of convection, while in the latter case of inertial-gravity waves we reduced it into a simplified theory of balanced flow adjustment, which give potential to study typhoon properties in section 6.

The mathematical study which is based on asymptotic expansions and looks at possible resonant interactions between gravity waves and the vortical motion; this is standard in this business and the authors have nothing new to offer.

Reply: Yes, method of asymptotic expansions is nothing new, but it is used to study triggering mechanism of convection, which is new. Let me correct your above saying of "resonant interactions between gravity waves and the vortical motion". It's not gravity waves, it's convection modes. Don't you think a study on triggering mechanism of convection is new enough topic ? Moreover, the triggering is related to the unbalanced nature of vortical flow, such as unsteady flow, vorticity advection and density advection, and all these can have potential application in meteorological study.

Moreover, I have serious doubts that the present work is of any use. The equation they use to built their theory, I quote, "is not closed". I don't see how someone can claim growing or decaying and balanced or imbalanced solutions for a non closed equation.

Reply: The equation itself is not closed, but it is derived from a closed set of equations, which means all these equation are satisfied simultaneously. So our equation have same balanced/imbalanced solution with the original closed equations. I suggest also that you can read Zhao et al (2011) or the present paper about what we defined by "apparent instability" for growing or decaying.

I find referee #2 of this paper has a very good description about this issue, he says in his comments "the equation is somewhat complicated but nonetheless amenable to analysis of its expected behaviours". I can make a further explanation about that as

below:

Since vorticity equation, divergence equation and thermodynamic equation are substituted into (1) in the derivation, constraints form basic dynamical and thermo-dynamical laws that one might expect in convective systems in the real atmosphere still work to a certain extent in (1). Nevertheless, equation (1) is more appropriately to be regarded as a diagnostic equation for the relationship between unbalanced basic flow and convection/IGWs. Although it alone is not closed and cannot serve as the governing equation to decide the motion, it does serve as one of the constraints for the motion. So, it may unable to describe all aspects of the motion, but it can describe qualitatively one aspect: relationship between unbalanced basic flow and convection/IGWs.

I'd like to give a simple example to explain what the difference between closed governing equations and a constraint (or diagnostic equation) is. The continue equation can be written as below

Here is the horizontal divergence, i.e.

If there is a motion with horizontal divergence , we cannot know vertical motion $\omega$ everywhere of any time, because (a) is not a closed governing equation, but as (a) is a constraint to the motion, we know there must be a convergent vertical motion with . This is also the case of our equation (1), which is a compound constraint of basic dynamical and thermo-dynamical laws, but is not closed governing equation. In typhoon study, it describes a imbalance forcing and balanced flow adjustment relationship, which can be explained mathematically as below. In typhoon study ( for issues we discussed, there are strong balance), we use following equation

it can be written identically as

if r is far from meso-scale region, $\delta n$ is nearly free wave, and dispersion of IGW demands $\delta n \rightarrow 0$ as t $\rightarrow \infty$. So the only way for that is R(r,t)$\rightarrow$0 as t $\rightarrow \infty$. So adjustment tends to remove imbalance. This is what we can infer from a constrain or a diagnostic

equation, without solving the closed governing equations.

Furthermore, the paper is poorly written and full of typos.

Reply: These can be improved (probably with the help of native English speaker).

For all these reasons, it must be rejected and I am reluctant to waste my time to write a detailed report to send to the authors or post online because it will counterproductive.

Reply: My only suggestion to editor is to consider comments of all reviewers of both the present paper and Zhao et al (2011).

Please also note the supplement to this comment:
http://www.nonlin-processes-geophys-discuss.net/npg-2016-6/npg-2016-6-AC1-supplement.pdf

---

## Author Comment (AC2) · 9 Mar 2016

Reply to referee #2

Thank you for your careful reading and insightful comments on our manuscript. At this stage, I just make comments on your comments about scientific issues. Other problem will be responded in the revision of the manuscript, as long as editor allow us to submit a final version.

General Comments [This submission is a follow up to Zhao et al (2011) and rests on Eq. 1 here, which was obtained in that earlier study. The equation is somewhat complicated but nonetheless amenable to analysis of its expected behaviours. That is

what is attempted here for most of the submission. Then in the final section, there is an attempt to apply this thinking to certain aspects of typhoons. The analysis is not always well explained or presented, and could probably be shortened by tightening it up. For example, as detailed below, although the various claims do appear quite reasonable, some things are not fully justified and are stated too strongly, with plausible assumptions being described almost as though they were actual proofs of properties of the equation. Much of the analysis is nonetheless acceptable, although not particularly enlightening in isolation. I found myself awaiting the application rather impatiently in order to see whether working through the analysis would prove worthwhile. Alas, this was not the case.]

Reply: Indeed it is too long, we can shorten it. Since you do not point out here where other problem are in the manuscript, I suppose I can just find them in your Specific Comments and respond. Also, I think what you called by "plausible assumptions" are something used quite often in meteorological community and are not our inventions, although they are not exact.

[The issue is that the authors offer no more than possible explanations of how a full set of calculations might be able to provide some explanation for properties of typhoons. Again, statements are over claimed. Often this to the effect that things have been explained when in fact the "explanation" has gone no further than speculations as to what aspects of the equation might be able to play what role in providing explanations.]

Reply: I don't think our explanation is something of "speculations as to what aspects of the equation might be able to play what role in providing explanations". The physical meaning is clear and sound: the process is balanced flow adjustment, a very classical concept, in which when a flow depart far from balanced flow such as geostrophic flow and axisymmetric gradient flow, it will adjust itself toward balanced flow by the emission and dispersion of IGW. The equations derived and used are based on such physical concept. We also show imbalance of vortical flow are related to unsteady flow, vorticity advection and density advection as well as their asymmetry. So process of balanced

flow adjustment tends to remove these imbalance, by which we can explanation properties of typhoons. So this part of work has strong physical background and cannot be simply regarded as some superficial mathematical speculations. Please consider it again. Also, if there are problem in reducing our equation into a simplified theory of balanced flow adjustment, could you please point out. If you feel statements are over claimed, we can state as a possible physical explanation. Again, I'd like to give further explanation of the physical meaning of eq (1). I think that adjustment process occurs on some condition of R, while instability occurs on other condition. There is a cycle :

Strong R -> adjustment->balance-> unstable atmosphere ->instibility -> imbalance-> strong R-> and so on . . .

So, the process with a reduction of R may be regarded as balanced flow adjustment process. All of our examples of typhoon properties are related with the process of reduction of R, so they can be regarded as balanced flow adjustment process,

[ Really the study is crying out for actual calculations to be done using the equations presented. I don't see that this would be an impractical task. Idealized simulations of typhoons could be used to provide suitable background data with which to solve the equations, and to test the authors' ideas and hypotheses properly. If that were to be done, there could ultimately be a strong paper that emerged, but it would very likely be a very different paper from the article submitted here.]

Indeed idealized simulations of typhoons is helpful for the understanding such issues. But this need a large amount of work and should appear in at least one more papers. Moreover, this part is not the whole story of our paper, we also deal with triggering mechanism of convection and how negative-sigma area affect convection/IGW modes. Also I must admit that I'm not sure whether or not I'm able to simulate successfully issues we discussed of typhoon properties, because I have a feeling that if numerical simulations really had been so effective, issues of typhoon's self-organization, Fujiwhara effect and the relationship between typhoon's asymmetric structure and its track

recurvature should have been fully understood. Obviously this is not the case. If that were to be done, we could publish it as other papers without further explanation like above in other journals of meteorology. So at this stage of preliminary study, we can give just "a possible physical explanation" for such issues and submit it to NPG which we think is primarily a journal for theoretical study.

Specific Comments [1. p5, line 14. The equations are stated in the Appendix but a summary of their properties is not given. It would have been useful to do so.]

Reply: We can do that in the revised version.

[2. p7, line 17, "it is proven" does not seem to be the right wording here. Rather the statement follows simply from what the authors mean by balanced or imbalanced in this context.]

Reply: We can change it in the revised version.

[3. Eq. 1. Much of the analysis in the paper ignores the nonlinear operator on _, specifically I(_) in Eq. 2b. However, this needs some more motivation and discussion in order to clarify the circumstances under which this will or will not be a reasonable assumption. On page 9, line 6 for instance we are told that this is done "for simplicity" and there is a little discussion at the end of the same section. However, the assumption seems to need more attention and should be stated much more clearly and strongly, with the caveats noted in the Introduction and Conclusions.]

Reply: This question was proposed also by a reviewer of Zhao et al (2011) and was addressed in the text of Zhao et al (2011). In the revised version of this paper we can addressed it in almost the same way and give the caveats noted in the Introduction and Conclusions. However, in the study of the triggering of convection, we do include this nonlinear term. And the linear equation afterward is obtained by the first order approximation of a perturbation method. So, in fact we have addressed this nonlinear term in this paper.

[4. The greek letter _ is not used consistently: see for example the different forms used on page 9, lines 20 and 21. This is not unusual and the authors need to go through the complete text to check.]

Reply: Thank for your careful reading, we will correct it.

[5. Eqs. 14 and 15. The quantities I2 and Ii need to be properly defined.]

Reply: We will give the definition in the revised version.

[6. Eq. 32. This assumption is introduced without comment. It is a reasonable approach to take in the analysis but it does need to be properly introduced, motivated and discussed, perhaps at the start of the Section.]

Reply: I see. We can change it in the revised version.

[7. p19, lines 18-19. At this stage it is far from clear that this statement about negative _ should hold true. Only later do we learn about the authors' arguments for it, and further that this statement is not necessarily true but simply a plasuible assumption. The argument is made at the end of page 20, and I have no complaint about it as a plausible assumption. But again it should not be stated as something stronger. It is unlikely but not out of the question that the area of negative _ may be rather large, or that the modulus of the An variations could be rather small. So the "inferred" for example, is not appropriate.]

Reply: Although I'm not sure whether we can observe area of sigma < 0, in almost all text books or papers of theoretical study on convective instability, the situation of sigma < 0 must be mentioned and discussed. Our work is not an exception. Maybe we can consider another case when there are several separate small areas of negative-sigma, I find (39) still holds and conclusion remains the same in this case. That is, convection modes are trapped in these small areas and IGWs are free. So, one single negative-sigma area may be small, but their ensemble seems very large. This seems more realistic.

[8. p20, line 15. Small relative to what, and with what justification?]

Reply: Small relative to short mode of inertial-gravity waves with c*k» f*f , c is wave speed, k is wavenumber.

[9. Eqs. 49 and 50. A comparison of these with Eq. 47 suggests an error somewhere here, given that cn is a dimensional quantity!]

Reply: I'll find what it may be.

[10. p29. The treatment of condensational heating needs much more discussion and motivation when it is introduced. Moreover it needs proper specification. Ascent is always staurated, but what about descent. The relation stated seems to imply latent cooling during descent which would be a strange assumption.]

Reply: I can understand this point. Nearly 30 year ago, people noticed theory of wave-CISK for MJO and other theory has such drawback, but can never be overcome so far. In fact, if we do not allow decent cooling, it becomes a very strong nonlinear function, a piecewise linear function, and cannot be dealt with. But in typhoon study, it is still used by many people, because the strongest vertical motion in a typhoon is ascent rather than decent. So there is strong heating in ascent, while there is only weak cooling in decent. This to some extent alleviates the difficulty.

Technical/Minor Corrections 1. p4, line 16, do not. 2. p7, line 19, remains should be singular. 3. p8, line 24, this should be reworded. 4. Eq. 7, it would be helpful to clarify here that the asterisk indicates an adjoint. In the current text, this only becomes clear a few pages later. 5. p13, line 14. Reword this sentence. 6. p14, line 11. quadratic. 7. p15, line 12, known should read shown. 8. Eq. 36b. the brackets and modulus signs need fixing. 9. p22, line 13, forced. 10. Some of the equation referencing seems to have gone awry: (a) p21, line 23. Eq. (24) is not right. 26? (b) p40, line 18. (c) p42, line 6 11. Eq. 48. The final t in the argument list for G should read t0 12. p25, line 10, complicated. 13. p25, line 17, and elsewhere. To avoid the obvious potential for

confusion the Rossby number should be denoted by the standard Ro and not by Re. C4 14. p27, line 20 and elsewhere. Fr is a poor choice of notation, since there could be scope for confusion with a Froude number. 15. p29, line 22, it is. 16. p31, line 9, parties should read parts. 17. p33, line 5, does not. 18. p35, line 16. Reword, the SST is not a heating. 19. p37, line 20, it is. 20. Eqs. B5, the n subscripts are missing.

---

## Short Comment (SC1) · 10 Mar 2016

Fujiwhara effect was initially found in the motion of vortices in water, where there are not precipitation or some kind of thermal forcing such as radiation and/or surface heat and/or moisture fluxes, but phenomenon in binary typhoon interaction is similar. This strongly suggest Fujiwhara effect of typhoons is due to purely dynamical process. Balanced flow adjustment is the dynamical process that we can suggest. Other physical processes in typhoon play just secondary roles in Fujiwhara effect. So, simulation by numerical model for Fujiwhara effect of typhoons may introduce something unnecessary.

---

## Short Comment (SC2) · 11 Mar 2016

The equation (1) itself is not closed, but it is derived from a closed set of equations, which means all these equation are satisfied simultaneously. So laws of conservation of vorticity and/or energy still hold. This is why balanced flow adjustment finally form one votex rather than static state or geostraphic flow which are also balanced flows.